# Bioactive C_17_ and C_18_ Acetylenic Oxylipins from Terrestrial Plants as Potential Lead Compounds for Anticancer Drug Development

**DOI:** 10.3390/molecules25112568

**Published:** 2020-05-31

**Authors:** Lars Porskjær Christensen

**Affiliations:** Department of Chemistry and Bioscience, Faculty of Engineering and Science, Aalborg University, Niels Bohrs Vej 8, 6700 Esbjerg, Denmark; lpch@bio.aau.dk; Tel.: +45-277-874-94

**Keywords:** acetylenic oxylipins, polyacetylenes, cytotoxicity, anti-inflammatory, lead compounds, mechanisms of action, structure-activity relationship, terrestrial plants

## Abstract

Bioactive C_17_ and C_18_ acetylenic oxylipins have shown to contribute to the cytotoxic, anti-inflammatory, and potential anticancer properties of terrestrial plants. These acetylenic oxylipins are widely distributed in plants belonging to the families Apiaceae, Araliaceae, and Asteraceae, and have shown to induce cell cycle arrest and/or apoptosis of cancer cells in vitro and to exert a chemopreventive effect on cancer development in vivo. The triple bond functionality of these oxylipins transform them into highly alkylating compounds being reactive to proteins and other biomolecules. This enables them to induce the formation of anti-inflammatory and cytoprotective phase 2 enzymes via activation of the Keap1–Nrf2 signaling pathway, inhibition of proinflammatory peptides and proteins, and/or induction of endoplasmic reticulum stress, which, to some extent, may explain their chemopreventive effects. In addition, these acetylenic oxylipins have shown to act as ligands for the nuclear receptor PPARγ, which play a central role in growth, differentiation, and apoptosis of cancer cells. Bioactive C_17_ and C_18_ acetylenic oxylipins appear, therefore, to constitute a group of promising lead compounds for the development of anticancer drugs. In this review, the cytotoxic, anti-inflammatory and anticancer effects of C_17_ and C_18_ acetylenic oxylipins from terrestrial plants are presented and their possible mechanisms of action and structural requirements for optimal cytotoxicity are discussed.

## 1. Introduction

Polyacetylenes are widely distributed in nature, occurring in terrestrial plants, fungi, bacteria, mosses, lichens, marine algae, and invertebrates. The majority of polyacetylenes are characterized by a long hydrocarbon chain consisting of 10 to 42 carbons, and among these, C_17_ and C_18_- polyacetylenes have often been reported to possess potential anticancer effects [1,2,3,4,5]. More than 2500 polyacetylenes are known, of which the majority have been isolated from terrestrial plants and, in particular, from the botanically related plant families Apiaceae, Araliaceae, and Asteraceae [1,3]. In plants, polyacetylenes are biosynthesized from unsaturated fatty acids such as oleic acid and linoleic acid by dehydrogenation leading to the C_18_ acetylenic oxylipins crepenynic acid and dehydrocrepenynic acid. These polyacetylenic precursors are then transformed into C_18_ acetylenic oxylipins by further dehydrogenation and oxidation reactions or they may undergo β-oxidation and/or α-oxidation leading to polyacetylenes of various chain lengths, thus polyacetylenes isolated from terrestrial plants have typical chain lengths ranging from 10 to 18 carbon atoms [1,2,3]. Many plants belonging to particular Apiaceae and Araliaceae families are known for their cytotoxic, anti-inflammatory, and potential anticancer properties. Although these pharmacological properties have often been ascribed to the presence of bioactive terpenoids such as sesquiterpene lactones in Apiaceae and Asteraceae, and triterpenoid saponins in Araliaceae, there is also strong evidence from numerous investigations that polyacetylenes constitute a group of highly bioactive compounds in plants of these families contributing to their pharmacological properties [5,6,7,8,9,10]. The polyacetylenes in the Apiaceae and Araliaceae are mainly aliphatic C_17_ and C_18_ acetylenic oxylipins, of which acetylenic oxylipins of the falcarinol-type are the most common, whereas in the Asteraceae a much larger structural diversity among polyacetylenes is observed, although falcarinol-type oxylipins rarely occur in this family [1,3,9,10,11,12]. However, in some tribes of the Asteraceae such as Anthemideae, Astereae, and Heliantheae, C_17_ acetylenic oxylipins of the dehydrofalcarinol-type are common [11,12,13], and, like the related falcarinol-type oxylipins, they possess interesting cytotoxic and anti-inflammatory activities [8].

The triple-bond functionality of acetylenic oxylipins seems to transform these fatty acid derivatives into highly alkylating compounds that are able to trap thiols or other nucleophiles by direct nucleophilic addition to their unsaturated electrophilic systems and thus bind covalently to proteins or other biomolecules with nucleophilic functionalities, as shown in Figure 1 for (3*R*)-falcarinol [14,15,16]. Acetylenic oxylipins of the falcarinol-type have, for example, shown to bind covalently to the efflux protein ABCG2 (also known as the breast cancer resistance protein (BCRP)) involved in breast cancer chemotherapy resistance [17] as well as to the cannabinoid receptor 1 (CB_1_) and GABA_A_ receptors resulting in modulation of the activity of these receptors [15,18]. In addition, this type of acetylenic oxylipins has also been shown to bind covalently to cysteine in enzymes such as mitochondrial aldehyde dehydrogenases (ALDHs) in cancer cells leading to a reduction of activity [14]. Reduction of the activity of ALDHs may lead to oxidative stress and endoplasmic reticulum (ER) stress causing cell injuries, cell cycle arrest, and apoptosis [19,20], and thus could be one of the mechanisms of action that could explain the cytotoxicity of falcarinol-type polyacetylenes as well as other acetylenic oxylipins. Furthermore, acetylenic oxylipins may act as ligands for nuclear receptors such as peroxisome-proliferator-activated receptors (PPARs). Besides being involved in lipid and glucose metabolism, the PPARs are also involved in cell proliferation and apoptosis [21,22]. For example PPARγ is involved in glucose metabolism through the improvement of insulin sensitivity, representing a potential therapeutic target of type 2 diabetes, but is also thought to have overall anticancer effects in many different cancer cell types, due to its antiproliferative and proapoptotic properties [21,22]. Endogenous ligands for the PPARs are fatty acids and fatty-acid derivatives and it has, for example, been demonstrated that (3*R*)-falcarinol (**1**) and (3*R*,8*S*)-falcarindiol (**5**) [Figure 2] ligand bind and activate PPARγ [23,24,25], which may to some extent explain their potential antidiabetic effects but also their anticancer effects.

The electrophilic properties of C_17_ and C_18_ acetylenic oxylipins and their reaction with thiols in cysteine indicate that they are able to activate the Kelch-like ECH-associated protein 1 (Keap1)/nuclear factor erythroid 2–related factor 2 (Nrf2)/antioxidant response element (ARE) pathway. The Keap1-Nrf2 pathway regulates the expression and formation of a battery of antioxidant, anti-inflammatory, and cytoprotective phase 2 enzymes and therefore this pathway is important in the chemopreventive protection against carcinogens and inflammation [16,26,27,28,29,30]. Nrf2 is a member of the NF-E2 family of the basic leucine zipper transcription factors, and is a key transcription factor. Nrf2 regulates the target genes related to detoxification and antioxidation by binding to ARE. In resting cells, Nrf2 is bound to Keap1; however, upon exposure to various stimuli, including reactive oxygen species and electrophilic molecules, Nrf2 is activated and released from the Keap1 complex and translocated to the nucleus to activate its target genes. The release of Nrf2 from Keap1 may also depend on phosphorylation of Nrf2, which is mediated through multiple kinases including ERK1/2, PKC, p38, and PKB (Akt) [28,29]. Carcinogens have been shown to activate the proinflammatory nuclear factor kappa-light-chain-enhancer of activated B cells (NF-κB) pathway [31]. Consequently, the activation of the Keap1-Nrf2 pathway by acetylenic oxylipins may contribute significantly to their chemopreventive effects through detoxification of carcinogens and increased production of anti-inflammatory molecules such as carbon monoxide and cytokines that are produced by enzymes such as heme oxygenase-1 (HO-1) [30].

Chronic inflammation is associated with an increased risk of cancer, being linked to approximately 25% of all human cancers [32]. Common causes of chronic inflammation associated with cancer development include *Helicobacter pylori* infections in gastric cancer, human papilloma virus in cervical cancer, hepatitis B or C infections in hepatocellular carcinoma, and inflammatory bowel disease in colorectal cancer (CRC) [33,34,35]. The transcription factors NF-κB and signal transducers and activators of transcription 3 (STAT3) are two major pathways of inflammation that are activated by, for example, infections that cause chronic inflammation, and thus these transcription factors play a central role in inflammation-induced cancers [33,35,36]. NF-κB mediate the expression of proinflammatory cytokines, such as tumor necrosis factor alpha (TNFα), interleukin (IL)-1, and IL6, as well as inflammatory enzymes, such as cyclooxygenase-2 (COX-2) and 5-lipooxygenase (5-LOX), which are all expressed in chronic inflamed tissue [33,36]. These proinflammatory stimuli promote carcinogenesis, forming a rich and complex network of inflammatory responses within the tumor microenvironment contributing to survival, proliferation, invasion, and metastasis of tumors. COX-2 levels are low in normal tissue but are rapidly induced as an early response to growth factors, cytokines and tumor promoters associated with inflammation, cell survival, abnormal proliferation, angiogenesis, invasion, and metastasis [37]. Thus COX-2 has an important function in driving carcinogenesis and this is done through the production of prostaglandins (PGs), which inhibit apoptosis and enhance cell migration of cancer cells, and promote the formation of blood vessels in tumor tissue (neoangiogenesis) [36,37,38]. COX-2 levels are increased in many types of tumors in colorectal [39], bladder [40], breast [41], lung [42], pancreas [43], prostate [44], and head and neck cancer [45], thus inhibition of COX-2 is an important target for anti-inflammatory drugs in the treatment of many cancers. TNF-α produced during chronic inflammation appears to enhance tumor development and dissemination as it is a major cytokine in the tumor microenvironment, being capable of regulating other proinflammatory cytokines and hence is able to influence several of the hallmarks of cancer, including stimulation of tumor-cell growth, survival, invasion, metastasis, and neoangiogenesis [46,47]. Drugs that inhibit TNF-α signaling in inflammatory conditions are therefore of great interest for the treatment of various cancers. IL-6 is another major tumor-promoting cytokine produced by both malignant and host cells within the tumor microenvironment [48]. Excess IL-6 production drives carcinogenesis and for some types of cancers high circulating levels of IL-6 indicate a poor prognosis [49,50]. Likewise, overexpression of COX-2 also indicates poor prognosis for several types of cancer [39,40,51].

Bioactive C_17_ and C_18_ acetylenic oxylipins have been shown to inhibit NF-κB and the formation of proinflammatory cytokines and inflammatory enzymes such as ILs, COXs and LOXs and, therefore, the direct inhibition of these inflammatory mediators appears to be another important mechanism of action for the prevention and treatment of cancer by these secondary metabolites. This has recently been demonstrated for (3*R*)-falcarinol and (3*R*,8*S*)-falcarindiol isolated from carrots in a rat model of CRC, where it was shown that these polyacetylenes selectively inhibited the expression of COX-2 in tumor tissue as well as TNF-α and IL-6, thus explaining the CRC preventive effect of carrots and related root vegetables [52,53]. This will be discussed in more details in Section 4.1. Although, no pharmacokinetic study in tumors and plasma in this rat study was performed, the lipophilic nature of these acetylenic oxylipins clearly indicate that they are absorbed in cells affecting tumor growth. This assumption is supported by a pharmacokinetic study of (3*S*,8*S*)-falcarindiol (**10**) and oplopandiol (**18**) in rats after oral administration of a polyacetylene extract of *Oplopanax elatus* Nakai demonstrating that these polyacetylenes were rapidly absorbed in vivo [54]. This is also in accordance with a human trial demonstrating that (3*R*)-falcarinol and (3*R*,8*S*)-falcarindiol is rapidly absorbed in humans after oral administration of carrot juice containing these polyacetylenes [9,55,56]. The preliminary bioavailability studies clearly show that C_17_ and C_18_ acetylenic oxylipins are bioavailable. This is important in relation to the interpretation of their cytotoxicity and anti-inflammatory activity in vitro to a chemopreventive effect in vivo, because bioavailability is a prerequisite for the latter, although in vitro experiments may not be able to predict accurately, the effects of these polyacetylenes in vivo. The anticancer effect of acetylenic oxylipins have only been investigated in a few animal studies as discussed in Section 4, thus the potential chemopreventive effect of these secondary metabolites are, with a few exceptions, based on in vitro studies.

The electrophilic nature of C_17_ and C_18_ acetylenic oxylipins appears to play a central role for their cytotoxic, anti-inflammatory and chemopreventive activities. Furthermore, the structural similarities of these acetylenic oxylipins to endogenous ligands of PPARs and their ability to activate these nuclear receptors may also contribute to their anticancer effect. Therefore, these secondary metabolites seem to constitute a group of promising lead compounds for the development of anticancer drugs for the prevention and possibly treatment of cancers. In this review, the cytotoxicity and anti-inflammatory activity in vitro of C_17_ and C_18_ acetylenic oxylipins isolated from terrestrial plants, including some synthetic enantiomers of naturally occurring acetylenic oxylipins are presented, as well as investigations of their in vivo anticancer effects. Furthermore, possible mechanisms of action and structural requirements for optimal cytotoxicity of these acetylenic oxylipins are discussed.

## 2. In Vitro Cytotoxicity of C_17_ and C_18_ Acetylenic Oxylipins and Structure Activity-Relationship

### 2.1. Cytotoxic C_17_ and C_18_ Acetylenic Oxylipins Isolated from Plants of the Araliaceae

#### 2.1.1. Cytotoxic C_17_ and C_18_ Acetylenic Oxylipins from Panax Species

Many plants of the genus *Panax* (Araliaceae) have been used in traditional medicine in Asia and in North America against various types of illnesses and diseases. *Panax ginseng* C.A. Meyer is the most famous of the *Panax* species and is also known as Korean ginseng or Asian ginseng. The roots of *P. ginseng* have been used as an herbal remedy in eastern Asia for more than 2000 years and is known for its possible chemopreventive effects [57,58,59]. The chemopreventive effects of *Panax* species have mainly been associated with the content of triterpenoid saponins (ginsenosides) [60] until the discovery of the potential anticancer activity of the petroleum ether extract from *P. ginseng* roots around 1980 demonstrating cytotoxic effects to murine leukemia and sarcoma cells [61]. Since then, the lipophilic part of this plant and other *Panax* species such as *P. quinquefolius* L. (American ginseng), *P. notoginseng* (Burkill) F.H. Chen (Chinese ginseng) and *P. stipuleanatus* Tsai and Feng have been investigated for cytotoxic compounds. This had led to the characterization of several cytotoxic acetylenic oxylipins of the falcarinol-type (**1**, **2**, **20**–**22**, **29**, **31**, **32**, **34**–**38**, **41**–**43**, **50**, **52**, **53**, Figure 2), panaxydiol-type (**57**, **58**, **60**, **63**, **68**, **69**, Figure 3), and dehydrofalcarinol-type (**77**, **78**, **80**, **81**, Figure 4) as well as the related acetylenic ginsenoyne J (**95**) and ginsenoyne I (**96**) (Figure 5).

The first interesting studies on the cytotoxic activity of *P. ginseng* polyacetylenes appeared between 1987–1990. These early studies demonstrated that panaxydol (**20**), 1-chloropanaxydiol (**35**), dihydropanaxacol (**52**), and panaxacol (**53**) [Figure 2], isolated from dried callus, were cytotoxic against murine leukemia L1210 cells and Yoshida sarcoma cells, resulting in a strong inhibition of the growth of the cells at 10 μg/mL [62,63,64]. Furthermore, the cytotoxic activity of falcarinol (**1**), panaxydol (**20**), and panaxytriol (**41**) (Figure 2) isolated from red *P. ginseng* roots were found to be highly cytotoxic against human and murine malignant cells showing the strongest cytotoxic activity towards human gastric adenocarcinoma (MK-1) cancer cells with an ED_50_ (median effective dose) of 0.027, 0.016, and 0.171 μg/mL, respectively, but interestingly relatively low cytotoxicity to human nonmalignant cell lines [65,66,67]. Other C_17_ acetylenic oxylipins (**2**, **22**, **29**, **38**, **42**, **43**, **58**, **77**, **78**, **80**, **81**, **95**, **96**, Figure 2, Figure 3, Figure 4 and Figure 5) isolated from *P. ginseng* roots have since been tested for their cytotoxic activities against varies human malignant and murine malignant cells [68,69]. In one study falcarinol, panaxydol, and panaxytriol showed potent cytotoxicity against the murine malignant cells DT, 3T3, and L1210 that were comparable with the chemotherapy drugs 5-fluorouracil and cisplatin used as positive controls [68]. The most potent of the tested acetylenic oxylipins in this study was, however, ginsenoyne A (**77**), which was even more potent towards the human malignant cell lines HeLa, T24, and MCF-7 compared to the positive controls [68]. In another study the acetylenic oxylipins **1**, **20**, **41**, **42**, **77**, and **81** were tested for their cytotoxicity against the human cancer cell lines A549, SK-OV-3, SK-MEL-2, and HCT-15 and falcarinol and ginsenoyne A were the most potent, with ED_50_ values of 2.38–6.04 μM and 2.14–5.95 μM, respectively [69]. Most investigations on the cytotoxicity and mechanisms of action of polyacetylenes isolated from *P. ginseng* have, however, been performed on panaxydol and panaxytriol.

The inhibitory effects of panaxydol, isolated from *P. ginseng*, on carcinogenesis and tumor growth have been demonstrated in numerous investigations on different types of cancer cells [70,71,72,73,74,75,76]. The possible mechanisms of action for the antiproliferative effects of this acetylenic oxylipin appears to involve induction of G_1_ phase cell cycle arrest and/or apoptosis of cancer cells. Panaxydol has been shown to inhibit the growth of SK-MEL-1 cells, a human malignant melanoma cell line, and to induce cell cycle arrest at G_1_ phase in a dose-dependent manner (0–20 μg/mL), with significant effects at 10 μg/mL. The induction of G_1_ phase cell cycle arrest by panaxydol in this study was shown to be associated by decreasing cyclin-dependent kinase 2 and upregulation of the cyclin-dependent inhibitor p27^kip1^ [71]. Induction of cell cycle arrest at G_1_ to S transition in a dose-dependent manner have also been demonstrated in the human hepatocarcinoma cell line HepG2 with significant effect already at 4 μM [72]. The antiproliferative effects of panaxydol of HepG2 cells was shown to be dose- and time-dependent with IC_50_ (half maximal inhibitory concentration) values of 12.9, 8.2, and 6.3 μM at 24, 48, and 72 h, respectively. Furthermore, it was demonstrated that panaxydol upregulated the cyclin-dependent inhibitor p21^Waf1/Cip1^ and downregulated inhibitors of differentiation/DNA binding proteins Id-1 and Id-2 that are involved in the cell cycle regulatory system, which may explain the observed induction of cell cycle arrest of HepG2 cells [72]. Panaxydol has also been shown to induce cell cycle arrest at G_1_ phase in non-small-cell lung cancer cells (A549 and NCI-H358), and, like in SK-MEL-1 cells, the cell cycle arrest was accompanied by a downregulation of cyclin-dependent kinases and an upregulation of p27^kip1^ as well as an upregulation of p21^Waf1/Cip1^ [73]. In addition, a downregulation of cyclins (D and E) was observed. Cyclins are proteins that control the progression of a cell through the cell cycle by activating cyclin-dependent kinases. Finally, panaxydol was shown to induce accumulation of intracellular Ca^2+^ in non-small-cell lung cancer cells, and this upregulation of Ca^2+^ was found to be closely related to the effect on G_1_ cell cycle arrest [73]. Panaxydol has also been shown to suppress cell proliferation in rat C6 glioma cells by induction of p27^kip1^ [74]. Thus, panaxydol may represent a promising lead compound for the development of drugs for the treatment of lung cancer and other cancers. Moreover, panaxydol has also shown to induce apoptosis in human leukemia T cell line (Jurkat), and in a human breast cancer cell line (MCF-7) in concentrations of 40 and 50 μg/mL, respectively. This apoptotic effect was also associated with a rapid increase in the cytoplasmic Ca^2+^ concentration activating NADPH oxidase via p38/JNK resulting in the induction of oxidative stress and activation of mitochondria-dependent apoptosis [75]. In another study, it was demonstrated that panaxydol induces apoptosis in MCF-7 cells by activating the epidermal growth factor receptor (EGFR) and phospholipase Cγ (PLCγ) at 50 μg/mL, although a lower concentration of panaxydol (20 μg/mL) induced the same signaling events but with slower kinetics [76]. Activation of EGFR and PLCγ is usually associated with tumor cell migration and tumor growth in many types of cancer, but, in this study, it was accompanied with Ca^2+^ release from ER and eventually in an increased mitochondrial Ca^2+^ concentration resulting in oxidative stress and ER stress leading to the observed apoptosis [76]. In addition, combined treatment with panaxydol and the anticancer drug cisplatin exhibited a synergistic cytotoxic effect in vitro. Cisplatin is known to induce oxidative stress, suggesting the possibility of combination therapy. Based on the current studies on the antiproliferative effects of panaxydol, it appears that cell cycle arrest by activating p21^Waf1/Cip1^ and p27^kip1^, and ER stress are the main mechanisms of action. ER stress can lead to apoptosis but may also result in cell cycle arrest by activating the tumor suppressor protein p53 [77,78], which also operates by inducing the expression of p21^Waf1/Cip1^ [78,79].

Panaxytriol has, in addition to the cytotoxicity described above, shown to inhibit mitochondrial respiration of a human breast carcinoma cell line (M25-SF) [80] and to enhance the cytotoxic activity of the chemotherapeutic agent mitomycin C against the human gastric carcinoma cell line, MK-1, in a synergistic manner [81]. Moreover, studies on the inhibitory effects of carcinogenesis and tumor growth of panaxytriol in vivo, as will be discussed in Section 4.2, and in vitro as well as its possible mechanisms of action have been investigated. In the study by Kim et al. [82], the cytotoxicity of panaxytriol towards a number of murine and human cancer cell lines was investigated, including P388D1 (mouse lymphoma), Jurkat (human lymphoma), U937 (human lymphoma), K562 (human leukemia), NIH/3T3 (mouse fibroblast), SNU-1 (human gastric carcinoma), SNU-C2A (human colon cancer), PC3 (human prostate cancer), and MCF-7 (human breast cancer), with IC_50_ values of 3.1, 11.1, 9.8, 10.8, 7.0, 29.7, 8.3, 19.1, and > 40 μg/mL, respectively, measured through MTT(3-(4,5-dimethylthiazol-2-yl)-2,5-diphenyltetrazolium bromide) assay. Paclitaxel (taxol) was used as a positive control with IC_50_ values ranging from 2.0 to > 40 μg/mL. The study furthermore showed that the DNA synthesis was strongly inhibited in the tumor cells tested and for two of the most sensitive cancer lines, P388D1 and SNU-C2A, this inhibitory effect of DNA synthesis was dose-dependent. IC_50_ values for DNA synthesis inhibition were 0.7 and 7.8 μg/mL for P388D1 and SNU-C2A, respectively. In addition, panaxytriol was shown to induce cell cycle arrest of P388D1 cells at the G_2_/M phase, where the proportion of cells in the G2/M phase of the cell cycle increased from 9% (untreated cells) to 26% and 48% after 24 and 36 h exposure to 5 μg/mL, respectively. Thus, panaxytriol appears to have both a dose- and time-dependent inhibitory effect on cell proliferation and that a likely mechanism of action is induction of cell cycle arrest [82].

The polyacetylenic profile of the roots of *P. quinquefolius* is very similar to that of *P. ginseng*, although *P. quinquefolius* contain other cytotoxic polyacetylenes that have so far not been found in *P. ginseng*. Cytotoxic polyacetylenes that have been isolated from the roots of *P. quiquefolius* include, besides falcarinol (**1**), panaxydol (**20**) and panaxytriol (**41**), also the acetylenic oxylipins **21**, **29**, **31**, **32**, **34**, **36**, **37**, **42**, **50**, **57**, and **63** (Figure 2 and Figure 3). These polyacetylenes have, so far, only been tested against murine leukemia L1210 cells where they have shown strong inhibitory activity with IC_50_ values of 0.1–1 μg/mL [83,84,85,86]. It is, however, interesting that the cytotoxic activity of C_17_-polyacetylenes against L1210 cells is approximately 20 times stronger than the corresponding C_14_-polyacetylenes with a terminal diyne moiety. This indicates that a terminal double bond in combination with a secondary allylic alcohol or ester group or ketone at C-3 is important for the cytotoxicity of these C_17_-polyacetylenes and/or simply that the chain length is important for the activity as demonstrated for synthetic symmetric aliphatic diacetylenes, where the cytotoxicity drops off dramatically as the chain length shortens [87]. Furthermore, the stereochemistry of C_17_-polyacetylenes from *P. quinquefolius* is important for their cytotoxicity as shown in a study by Satoh et al. [83], who investigated the cytotoxicity of (3*R*,9*R*,10*S*)-panaxydol (**20**), (3*R*,9*R*,10*S*)-acetylpanaxydol (**21**), (9*R*,10*S*)-ginsenoyne E (**29**), (3*R*,9*R*,10*R*)-panaxytriol (**41**), (3*R*,9*R*,10*R*)-panaquinquecol 1 (**42**), (9*R*,10*R*)-3-oxo-panaquinquecol 1 (**50**), and (3*R*,10*S*)-panaxydiol (**57**) against L1210 cells as well as their synthesized optical isomers (**23**–**28**, **30**, **44**–**49**, **51**, **59**–**61**) [Figure 2 and Figure 3], of which a few also occur naturally in plants of the Araliaceae family. In this study it was demonstrated that the (3*S*)-isomers of panaxydol (**23**, **27**), acetylpanaxydol (**24**, **28**), panaxytriol (**44**, **48**), panaquinquecol 1 (**45**, **49**), and panaxydiol (**59**, **61**) were approximately 10 times more potent with IC_50_ values of 0.01–0.1 μg/mL compared to those with (3*R*)-configuration (**20**, **21**, **25**, **26**, **41**, **42**, **46**, **47**, **57**, **60**) with IC_50_ values of 0.01–0.1 μg/mL [83]. The epoxide group in panaxydol, acetylpanaxydol, and 3-oxo-panaquinquecol 1 is, like the diyne functionality, susceptible to nucleophilic attack, and, therefore, this functional group is expected to have some effect on the cytotoxic activity of these compounds. However, no difference in activity between the (9*R*,10*S*)- and (9*S*,10*R*)-isomers was observed, which indicates that the epoxide group does not enhance cytotoxicity significantly compared to the corresponding dioxygenated polyacetylenes and that the stereochemistry at C-9 and C-10 is of less importance for the activity of C_17_ acetylenic oxylipins. Furthermore, the 3-oxo-acetylenic oxylipins, i.e., **29**, **30**, **50**, and **51** showed almost the same inhibitory activities as the corresponding (3*R*)-isomers (**20**, **25**, **42**, **47**). The above results indicate that the diyne functionality and the stereochemistry at C-3 are most important for the cytotoxic activity of C_17_ acetylenic oxylipins isolated from *P. quinquefolius*.

*Panax notoginseng* has not been studied as thoroughly for its content of cytotoxic polyacetylenes as *P. ginseng* and *P. quinquefolius*. However, the major polyacetylenes in *P. notoginseng* are the same, i.e., falcarinol (**1**), panaxydol and panaxydiol [88,89]. Falcarinol and panaxydol isolated from *P. notoginseng* have been investigated for their inhibitory effects on cell growth of human acute promyelocytic leukemia (HL-60) cells. It was found that they markedly inhibited proliferation of HL-60 cells in a time- and dose-dependent manner via an apoptotic pathway associated with proteolytic cleavage of the protein kinase C delta type (PKCδ), caspase-3 activation and degradation of poly ADP ribose polymerase (PARP) [89].

*Panax stipuleanatus* is a traditional herb that grows mainly in the north of Vietnam and the roots have been used in traditional medicine for the treatment of bleeding and muscular pain. Recent research has also indicated that this herb has anticancer and anti-inflammatory activities [90]. Chemical investigations of the roots of *P. stipuleanatus* resulted in the isolation of the cytotoxic panaxydiol-type polyacetylenes **60**, **68** and **69** (absolute configuration at C-3 and C-10 in this case unknown) [Figure 3] as well as panaxytriol (**41**) [90,91]. Among them, (3*R*,10*R*)-panaxydiol (**60**) and stipuol (**68**) showed significant cytotoxic activity, with IC_50_ values of 0.13 and 0.28 μM, respectively, against HL-60 cells, and 0.50 and 0.80 μM, respectively, against human colon cancer (HCT-116) cells. Investigation of the mechanisms of action of these polyacetylenes indicated that they markedly inhibited the proliferation of HL-60 and HCT-116 cells by induction of apoptosis [90].

#### 2.1.2. Cytotoxic C_17_ and C_18_ Acetylenic Oxylipins from Other Plant Species of the Araliaceae

Other medicinal plants of the Araliaceae family that have resulted in the isolation of cytotoxic acetylenic oxylipins include species from the genera *Aralia*, *Dendropanax*, *Hedera*, *Oplopanax*, and *Schefflera*. *Oplopanax horridus* (Araliaceae), known as Devil’s Club, is probably the most important ethnobotanical to indigenous people living in the Pacific northwest of North America, and, except for some *Panax* species, it is perhaps one of the most studied medicinal plants of the Araliaceae family. The traditional medicine uses the stem or root bark of *O. horridus* in treatment of external and internal infections and illnesses such as arthritis, respiratory ailments, digestive tract ailments, and fever [6,92,93,94]. In addition, it has been described that indigenous people in North America have used infusions of the bark of *O. horridus* as a possible treatment of cancer, although no solid documentation of its traditional use in cancer treatment exits [6]. However, recent pharmacological research has shown that the hydrophobic parts of ethanol extracts of root bark extracts of *O. horridus* inhibit the proliferation of several breast, colorectal, lung, ovarian, and acute myeloid leukemia cancer cell lines as well as cancer cells in animal models [6,94,95,96,97,98,99,100]. Investigations of *O. horridus* extracts, in particular in colon cancer cell lines, have shown that the mechanisms of action for the anticancer effect of the extracts is most likely caused by induction of apoptosis and arrest of the cell cycle at the G2/M phase [6,94,95,97,99]. The bioactive constituents responsible for the possible anticancer effect of *O. horridus* extracts appears mainly to be due to the C_17_ and C_18_ acetylenic oxylipins (3*S*)-falcarinol (**3**), (3*S*,8*S*)-falcarindiol (**10**), oplopandiol (**18**), oplopantriol A (**97**), (9*Z*,11*S*,16*S*)-11,16-dihydroxyoctadeca-9,17-dien-12,14-diyn-1-yl acetate (**98**), oplopantriol B (**100**), and oplopandiol acetate (**101**) [Figure 2 and Figure 5]. These polyacetylenes have all been isolated from root and/or stem bark extracts of *O. horridus* [6,93,94,95,101,102,103,104,105,106]. Investigations of the antiproliferative effects of the polyacetylenes from *O. horridus* have mainly been performed on human breast cancer (MCF-7, MDA-MB-231), human colon cancer (HCT-116, HT-29, SW-480), human liver cancer (HepG2), human pancreatic cancer (PANC-1, Bx-PC3) and human lung adenocarcinoma epithelial (A549) cell lines [6,94,102,103,104,105,106,107]. Based on these in vitro studies, it appears that (3*S*,8*S*)-falcarindiol (**10**) and oplopantriol A (**97**) show the strongest antiproliferative effects, in particular towards colon cancer cell lines, where they have been shown to significantly inhibit the cell growth of HCT-116, HT-29, and SW-480 cell lines at or below 10 μM. Their mechanisms of action appear to be cell cycle arrest in the G2/M phase and inhibition of proliferation by the induction of apoptosis at both earlier and later stages; thus contributing to an explanation for the observed anticancer activity of *O. horridus* extracts [94,102,103]. In the case of oplopantriol A, it has furthermore been demonstrated that this cytotoxic compound suppresses growth of HCT-116 and SW-480 cells both in a concentration- and time-dependent manner, with IC_50_ values of approximately 5 μM for HCT-116 and 7 μM for SW-480 cells [103]. Moreover, it has been shown that oplopantriol A inhibits cell proliferation and induces cell death in breast cancer (MDA-MB-231), mouse leukemia (MLL-AF9), HCT-116 as well as in lymphoblastic leukemia (KOPN1) cells but not in nontumorigenic cells, indicating that oplopantriol A preferentially is cytotoxic towards cancer cells [106]. The mechanisms of action by which oplopantriol A induces cell death in these cancer cells has been shown to be through induction of endoplasmic reticulum (ER) stress and the apoptotic BH3 proteins Noxa and Bim [106]. More or less the same effects have been observed for falcarindiol (**10**) on HCT-116 and SW-480 cells where it has been demonstrated that falcarindiol preferentially induced cell death and apoptosis of cancer cells but not normal colon epithelial cells and that its antiproliferative effect was mediated by induction of ER stress and activation of the unfolded protein response (UPR) [107]. Furthermore, the antitumor effects of falcarindiol and oplopantriol A have been demonstrated in xenograft tumor models in mice, as discussed in Section 4.2. Thus, ER stress and, thereby, induction of apoptosis and cell cycle arrest could be an important mechanism of action for the cytotoxicity of these acetylenic oxylipins in cancer cells.

A study on the cytotoxic effect of synthetic falcarinol (mixture of isomers), falcarindiol 3-acetate (**8**, mixture of isomers), and dihydrofalcarindiol (**17**), as well as the C_18_ derivative of oplopantriol A (**98**), isolated from *O. horridus*, against human pancreatic ductal adenocarcinoma cell lines PANC-1 and BxPC-3 showed that synthetic falcarinol, falcarindiol 3-acetate, and the C_18_ polyacetylene **98** were potent inhibitors of proliferation with IC_50_ values of < 1 μg/mL. In comparison, the synthetic dihydrofalcarinol gave IC_50_ values of > 10 mg/mL when tested against PANC-1 and Bx-PC3 cells [108]. Studies on the mechanisms of action of falcarinol and the C_18_ polyacetylene **98** showed that both compounds modulated genes related to proapoptosis, antiapoptosis, apoptosis, cell cycle, stress, and death receptors and is more or less in accordance with the mechanisms of action of these compounds on other types of cancer cells [108]. This study indicates that C_17_ and C_18_ acetylenic oxylipins with a terminal double bond are potent inhibitors of pancreatic cancer cell proliferation compared to corresponding polyacetylenes with a terminal single bond. However, from a study on the cytotoxicity of two falcarinol-type polyacetylenes, oploxyne A (**39**) and oploxyne B (**54**), with a terminal single bond and isolated from the related *Oplopanax elatus* [109,110], it is clear that the cytotoxicity also depends on the cancer cell lines. The screening of oploxyne A and B and a synthetic epimer of oploxyne A (**40**) for cytotoxicity against the cancer cell lines A549 (lung cancer), MCF-7 (breast), DU-145 (prostate), and SK-N-SH (neuro-blastoma), measured through a MTT assay, showed that oploxyne A (IC_50_ value of 7 μM) and its C-10 epimer (**40**) (IC_50_ value of 12 μM) were better than or similar to the positive control doxorubicin (IC_50_ value of 9 μM) against the human neuroblastoma cell line. Oploxyne B, on the other hand, was effective against DU-145 cells with an IC_50_ value of 17 μM but less effective than doxorubicin (IC_50_ value of 11 μM). However, against the other cancer cell lines, oploxyne A and its epimer showed moderate or no cytotoxic effect (IC_50_ > 27 μM) and the same is true for oploxyne B (IC_50_ > 30 μM) [110].

Based on a structure–activity relationship analysis of the cytotoxic activity of C_17_ and C_18_ acetylenic oxylipins, isolated from *O. horridus*, and synthetic derivatives, it appears that a terminal double bond (vinyl group) is important for their cytotoxicity as they seem to be much more cytotoxic than their corresponding dihydro derivatives, i.e., **17**, **18**, **100** and **101** [6,94,102,104,108]. This observation could indicate that this type of terminal double bond makes polyacetylenes more reactive and/or easier to transform into other metabolites in vivo. This may result in an induction of phase II enzymes and other cytoprotective enzymes such as NAD(P)H: quinone: oxidoreductase (NQO-1) and HO-1 as demonstrated for falcarindiol isolated from different Apiaceae plants [16,27,111]. This assumption is supported by data from the pharmacokinetic study of falcarindiol (**10**) and oplopandiol (**18**) in rats, which showed that the elimination half-life (T_1/2_) value of **18** was approximately 10 times higher than that of **10**, indicating a much higher metabolization rate in vivo of polyacetylenes with a terminal double bond compared to their dihydro derivatives [54]. Therefore, not only does the diyne system of C_17_ and C_18_ acetylenic oxylipins seem to play a role for the activation of the Keap1-Nrf2 pathway but so does the presence of a terminal double bond in the vicinity to the butadiynylcarbinol moiety of these acetylenic oxylipins. However, further studies are needed to explore the relationship and importance of such a terminal double bond with the activation of the Keap1-Nrf2 pathway and anticancer effect of C_17_ and C_18_ polyacetylenes. In addition, it has been shown that the cytotoxicity of synthetic acetylated derivatives of the polyacetylenes isolated from *O. horridus* become weaker if their hydroxyl groups are acetylated or they contain one more methylene group in the main skeleton chain, i.e., C_17_ polyacetylenes are, in general, more cytotoxic than the C_18_ polyacetylenes [104]. The cytotoxicity of the synthetic derivatives of *O. horridus* polyacetylenes are not included due to poor activity but a full description of their cytotoxicity can be found in reference [104]. Thus, the hydroxyl groups and the chain length of *O. horridus* polyacetylenes also appears to have some effect on their potential anticancer activity, and, furthermore, a terminal allylic secondary alcohol moiety contributes to the enhancement of the anticancer activity of C_17_ and C_18_ acetylenic oxylipins.

That a terminal double bond may be important for the cytotoxicity of C_17_ and C_18_ acetylenic oxylipins has been confirmed in a study on the cytotoxicity of falcarinol-type (**3**, **10**), panaxydiol-type (**59** or **61**, **67**) and dehydrofalcarinol-type (**72**, **73**) oxylipins isolated from *Dendropanax arboreus* (L.) Decne. & Planch. (Araliaceae) against the human disease-oriented tumor cell line screening panel (National Cancer Institute, USA) consisting of 60 human tumor cell lines (NCI-60 human tumor cell panel) [112]. The patterns of differential cytotoxicity for the isolated polyacetylenes from *D. arboreous* were virtually superimposable, although small differences in potency were observed through the series. Inclusion of a vinyl group in position 16,17 led to a slight reduction in potency from **3** → **72**, **10** → **73**, and **59**/**61** → **67**. The lowest cytotoxic potency was observed for dendroarboreol A (**79**), which is probably due to saturation of the vinyl group in position 1,2, in accordance with the structure-activity analysis of cytotoxic polyacetylenes isolated from *O. horridus*.

From Japanese ivy (*Hedera rhombea* Siebold and Zucc. ex Bean), an indole derivative of falcarindiol (**11**) has been isolated from the flower buds, which has shown interesting antiproliferative effects in the human promyelocytic leukemia HL-60 cell line [113,114]. This polyacetylene inhibited the HL-60 cell growth at 0.1 and 1.0 μg/mL, whereas it was cytotoxic at 10 μg/mL. The growth suppression induced by **11** was accompanied by G_0_/G_1_ phase arrest in the cell cycle at 1.0 μg/mL and, interestingly, induced the granulocytic differentiation of HL-60 cells. A common goal of cancer therapy is to restore normal growth of transformed cells, and therefore this falcarindiol derivative could act as a potential chemotherapeutic agent in human leukemia therapy.

Cytotoxic C_17_ and C_18_ acetylenic oxylipins in the Araliaceae, besides the above-mentioned plant species, have also been found in the roots of *Acanthopanax senticocus* (Rupr. et Maxim) Harms (**20**, **33**, **63**) where they seem to induce various proapoptosis mechanisms in animal cells [8]. From the leaves of *Schefflera taiwaniana* (Nakai) Kaneh., falcarindiol (**10**) and the related C_18_ polyacetylene **98** have been isolated showing inhibition of human nasopharyngeal carcinoma (HONE-1) and human gastric cancer (NUGC) cell lines at 50 μg/mL [115].

### 2.2. Cytotoxic C_17_ and C_18_ Acetylenic Oxylipins Isolated from Plants of the Asteraceae

Acetylenic oxylipins are characteristic for the Asteraceae and the large structural variation among this type of secondary metabolites in this family makes them appropriate chemotaxonomic markers. The chemical structures of these polyacetylenes also indicates that they are bioactive compounds with potential anticancer effects. Although investigations on the cytotoxicity of polyacetylenes isolated from Asteraceae are limited, it has been demonstrated that different types of polyacetylenes from this family possess cytotoxic activity including C_13_ and C_14_ spiro ethers, aromatic and aliphatic C_10_ and C_11_ polyacetylenes as well as C_17_ polyacetylenes of the dehydrofalcarinol-type [3,5,8]. Dehydrofalcarinol and its derivatives are common in certain tribes of the Asteraceae, but their potential anticancer effects have only been investigated in a few studies.

From an aqueous ethanol extract of the roots of *Gymnaster koraiensis* (Nakai) Kitamura, several cytotoxic polyacetylenes of the dehydrofalcarinol-type were isolated (**73**, **75**, **76**, **82**, **83**, **85**, **86**) by bioassay-guided fractionation using the murine leukemia L1210 tumor cell line as a model for cytotoxicity [116]. The chemical structure of gymnasterkoreayne E (**83**) in this study was later revised after the isolation of gymnasterkoreayne G (**84**) from *G. koraiensis* [117]. Of the cytotoxic C_17_-polyacetylenes from *G. koraiensis*, (3*S*,8*S*)-dehydrofalcarindiol (**73**) and gymnasterkoreayne C (**75**) exhibited significant cytotoxicity against the L1210 tumor cells, with ED_50_ values of 2.1 and 0.12 μg/mL, respectively, while compounds **76**, **82**, **83**, **85** and **86** were less potently cytotoxic with ED_50_ values of 3.1–10.4 μg/mL [116]. These cytotoxic polyacetylenes also occur in the flowers of the plant [118]. The above results indicates that a terminal double bond (vinyl group) in the vicinity to the diyne moiety is important for the enhancement of the cytotoxicity of *G. koraiensis* polyacetylenes against L1210 tumor cells, and confirm the results from the structure-activity analysis of other cytotoxic C_17_ acetylenic oxylipins described in Section 2.1. In addition, the cytotoxicity of (8*S*)-gymnasterkoreayne F (**85**) and its synthetic (8*R*)-isomer (**87**) have been evaluated over a five-log dose range in the NCI-60 human tumor cell panel exhibiting modest cytotoxicity with full-panel mean-graph midpoint (MG–MID) Log GI_50_ values of –4.44 (**85**) and –4.27 (**87**), respectively [119]. Nevertheless, gymnasterkoreayne F (**85**) had a Log GI_50_ value of –5.44 against the HL-60 leukemia cell line, whereas **87** had a Log GI_50_ value of –4.95 [119]. These results indicate that the naturally occurring (8*S*)-isomer of gymnasterkoreayne F is more cytotoxic than its synthetic (8*R*)-isomer, and these results therefore follow the tendency observed for the cytotoxicity of *Panax* polyacetylenes that the stereochemistry of chiral centers may be important for the cytotoxicity of C_17_ acetylenic oxylipins. Furthermore, it has been demonstrated that gymnasterkoreayne B (**82**) has chemopreventive properties and hepaprotective effects through induction of phase II detoxification enzymes NQO-1, HO-1, and glutathione reductase, as shown in mouse and human liver cells (HepG2) [120]. The cancer chemopreventive effects of **82** have also been investigated in HCT-116 human CRC cells and it was found that **82** significantly increases expression levels of Nrf2 and thus induces Nrf2 to activate phase II enzymes such as NQO-1 through the regulation of ERK and PKC pathways [121]. A structure-activity relationship study on the cancer chemopreventive effect of gymnasterkoreayne G (**84**) and synthetic diyne triol derivatives revealed that the diyne moiety is essential for the activity of these compounds [122], in accordance with the reactivity of the diyne moiety towards biological nucleophiles (Figure 1). Furthermore, this study also showed that the chain length has a strong effect on the chemopreventive activity of these compounds. Long chain lengths induced better NQO-1 activity, whereas, in terms of cytotoxicity, polyacetylenes with medium-sized or bulkier side chains exhibited better profiles [122]. However, most of the synthetic derivatives of gymnasterkoreayne G investigated in this study had chain length less than 17 carbon atoms and are therefore not included in this review but a full description of their chemopreventive activity can be found in reference [122].

Phytochemical investigations of *Cirsium japonicum* DC have resulted in the isolation of several cytotoxic dehydrofalcarinol-type polyacetylenes that includes 9,10-epoxy-heptadeca-16-en-4,6-diyn-8-ol (**88**), heptadeca-1-en,-l,13,-diyn-8,9,10-trio1 (**89**), and ciryneol A–C (**90**–**92**), which all inhibited the growth of KB cells (subline of HeLa cells) with IC_50_ values from 8.6 to 39.5 µg/mL [123,124].

From species belonging to the genus *Artemisia* several aromatic cytotoxic polyacetylenes have been isolated [3,8]; however, C_17_ polyacetylenes of the dehydrofalcarindiol-type are also common in this genus. From *Artemisia monosperma* Del. (3*R*,8*R*)-dehydrofalcarindiol (**74**) has been isolated and its cytotoxicity evaluated against a panel of colon cancer cell lines (LS174T, SKCO-1, COLO-320DM, WIDR) and breast cancer cell lines (MDA-231, MCF-7) with IC_50_ values of 9.6–14.8 μg/mL for colon cancer cells and IC_50_ values of 5.8 μg/mL and 37.6 μg/mL for MCF-7 and MDA-231 cells, respectively [125].

### 2.3. Cytotoxic C_17_ and C_18_ Acetylenic Oxylipins Isolated from Plants of the Apiaceae

#### 2.3.1. Cytotoxic C_17_ and C_18_ Acetylenic Oxylipins from Apicaceae Food Plants

Epidemiological studies have shown that vegetables may have cancer-preventive effects, including those found among Apiaceae food plants such as carrot, celery, celeriac, fennel, parsley, and parsnip, where cytotoxic acetylenic oxylipins of the falcarinol-type are common and thus may contribute to the potential cancer-preventive effects of these vegetables [7,9]. One of the most important apiaceous vegetables are carrots (*Daucus carota* L.), for two main reasons. First, they are consumed worldwide, in particular in North America and in European countries, and second, several meta-analysis studies on carrot consumption have indicated that carrots play a central role as a protecting vegetable, against development of different types of cancers [126,127,128,129]. The latter has recently been confirmed in a prospective cohort study, examining the risk of being diagnosed with CRC, as predicted by intake of carrots in a Danish population of 57,053 individuals with a long follow-up [130]. Self-reported intake of raw carrots at a baseline of 2–4 carrots or more each week (> 32 g/day) was associated with a 17% decrease in risk of CRC with a mean follow-up of > 18 years, compared to individuals with no intake of raw carrots [130]. The results of this prospective cohort study clearly support the results of the antineoplastic effects observed for the major polyacetylenic constituents in carrots in a rat model of CRC, as discussed in Section 4.1.

Carrots have been intensively investigated for their content of polyacetylenes and, so far, approximately 16 different C_17_ acetylenic oxylipins have been isolated from carrots of which the majority are of the falcarinol-type [7,9,131]. However, it is primarily the major polyacetylenes in carrots that have been investigated for their cytotoxic activity, i.e., (3*R*)-falcarinol (**1**), (3*R*,8*S*)-falcarindiol (**5**), and falcarindiol 3-acetate (**8**) [132,133,134,135,136,137]. The first study on the antiproliferative effects of falcarinol isolated from carrots appeared in 2003 and demonstrated that this acetylenic oxylipin stimulated differentiation of primary mammalian cells in low concentrations between 0.004 and 0.4 μM falcarinol, whereas cytotoxic effects were found above > 4 μM falcarinol [132]. This biphasic effect on cell proliferation is known as hormesis and is a characteristic feature of bioactive/toxic compounds [138,139], and has been demonstrated not only for falcarinol but also for falcarindiol and falcarindiol 3-acetate in different cell types of normal and cancer origin [134,135,136,137]. In human epithelial colorectal adenocarcinoma (Caco-2) cells, it has been shown that cell proliferation increased at relatively low concentrations of falcarinol (between 0.5–10 μM) where the expression of the apoptosis indicator caspase-3 decreased concomitantly with decreased basal DNA strand breakage. At concentrations above 20 μM falcarinol, proliferation of Caco-2 cells decreased and the number of cells expressing active caspase-3 increased simultaneously. Furthermore, DNA single-strand breakage was significantly increased at concentrations > 10 μM falcarinol [134]. Thus, the effects of falcarinol on proliferation of Caco-2 cells appears to be biphasic, inducing proproliferative and apoptotic characteristics at low and high concentrations of falcarinol, respectively. A biphasic effect on Caco-2 cell proliferation of falcarinol and falcarindiol has also been demonstrated in another study in concentrations ranging from 1 ng/mL to 20 μg/mL [136]. The same biphasic effects on cell proliferation have also been observed for falcarinol and falcarindiol in myotube cultures [135] and for falcarindiol 3-acetate in different leukemia cell lines [137]. Although a biphasic effect is a hallmark for many bioactive compounds, it is not possible based on the above investigations to conclude, whether the proproliferative effects observed at low concentrations of falcarinol-type polyacetylenes eventually will result in apoptosis or induce other cell responses that later may result in cell death.

From a number of in vitro studies, it appears that falcarinol (**1**) is usually more cytotoxic than falcarindiol (**5**) but that their cytotoxic potency against cancer cells depends on the cell lines [9,133,136,137,140,141]. Furthermore, it appears that falcarinol and falcarindiol may have a synergistic inhibitory effect on cell proliferation. In a study by Purup et al. [136], it was demonstrated that the cytotoxicity of lipophilic extracts from different carrot cultivars depended on the amounts of falcarinol, falcarindiol, and falcarindiol 3-acetate in the extracts. Extracts containing the highest concentration of falcarinol tended to have the highest growth inhibitory effect on Caco-2 cells, in accordance with a higher cytotoxic potency of falcarinol compared to falcarindiol. Moreover, it was shown that the cytotoxic effect on Caco-2 cells of falcarinol was enhanced synergistically when combined with falcarindiol in ratios of 1:1, 1:5 or 1:10 (Table 1). In addition, oxidation of the hydroxyl group at C-3 in falcarinol to falcarinone (**4**) resulted in a significantly less growth inhibitory effect in intestinal cells of both normal and cancer origin compared to falcarinol [136]. These results indicate that the cytotoxicity of some C_17_ and C_18_ acetylenic oxylipins may be enhanced when combined and that a hydroxyl group at C-3 may be important for their activity. The latter has also led to the proposition of an alternative alkylation reaction mechanism of falcarinol-type polyacetylenes involving the formation of a reactive resonance stabilized carbocation by the loss of water [136].

Moreover, falcarindiol and falcarindiol 3-acetate isolated from carrots have also shown to induce apoptosis in different leukemia cell lines (CCRF-CEM, Jurkat and MOLT-3) in concentrations from 18–68 μM and 23–38 μM, respectively, whereas falcarinol only caused induction of apoptosis in one of the cell lines (CCRF-CEM) at 45 μM [137]. On the other hand falcarinol was most cytotoxic towards the leukemia cells with IC_50_ values of 12–35 μM [137]. Falcarinol and falcarindiol have also shown in other studies that they can lead to cell cycle arrest and apoptosis of cancer cells, which may be linked to their alkylating properties as described in the introduction [94,142,143,144]. For example, it has been demonstrated that falcarindiol is able to induce ER stress in breast cancer cells (MDA-MB-231, MDA-MB-468 and SKBR3) leading to caspase-dependent cell death (apoptosis). In addition, falcarindiol contributed to autophagy-dependent cell death in these breast cancer cells and had synergistic effect with approved cancer drugs 5-fluorouracil and bortezomib in killing breast cancer cells [143]. Finally, it has been shown that falcarindiol may inhibit the growth of cancer stem cells by suppressing the Notch pathway, as demonstrated in neural stem cells [144]. The Notch pathway is vital to tumorigenicity of cancer stem cells, which are the driving force of tumor development [145].

A synergistic effect between polyacetylenes isolated from carrots may be important in order to explain the cancer-preventive effects of vegetables containing these compounds, but also indicates that they may have a slight different mechanism of action for their cytotoxic effects. This is in accordance with the different anti-inflammatory effects observed for falcarinol and falcarindiol (See Section 3) and from a recent study in a rat model of CRC, where it appears that their anti-inflammatory activity is enhanced in combination, acting as selective COX-2 inhibitors as described in Section 4.1. Furthermore, in vitro studies have shown that falcarinol and falcarindiol have different PPARγ activity; hence, not only do their alkylating properties appear to be important for their cytotoxic and anti-inflammatory activity but their ability to activate nuclear receptors such as PPARγ does also, which will be discussed in more details in Section 3 and Section 4. In addition, it has been shown that the dietary polyacetylenes falcarinol, falcarindiol, falcarindiol 3-acetate, and falcaridiol 3,8-diacetate (**9**) are inhibitors of the efflux protein ABCG2 [17]. ABCG2 is an efflux transporter that is expressed both in normal and in tumor cells and is important for xenobiotic absorption and disposition, and may play a role in multidrug resistance in cancer, although this has not been fully established [146]. The above-mentioned dietary falcarinol-type polyacetylenes inhibited the efflux of the ABCG2 substrate mitoxantrone in ABCG2-overexpressed human embryonic kidney 293 cells (HEK-293). This inhibitory effect of ABCG2 was furthermore confirmed in the vesicular transport assay, in which concentration-dependent inhibition of the uptake of the ABCG2 substrate methotrexate into ABCG2-overexpressed Sf9 membrane vesicles was observed with IC_50_ values of 19.7–41.7 μM [17]. Thus, the inhibition of ABCG2 by dietary falcarinol-type polyacetylenes may indicate a prospective use of these polyacetylenes as multidrug resistance reversal agents and thus a role in chemotherapy treatments.

From celery (*Apium graveolens*), the four polyacetylenes falcarinol, falcarindiol, 8-*O*-methylfalcarindiol (**6**) and (3*R*,10*S*)-panaxydiol (**57**) were isolated and tested for their cytotoxicity against an acute lymphoblastic leukemia cell line (CEM-C7H2), a human histiocytic lymphoma cell line (U937), a human multiple myeloma cell line (RPMI-8226), and the CRC cell lines HRT-18 and HT-2912 [133]. Falcarinol proved to be the most active of these polyacetylenes with a pronounced toxicity against CEM-C7H2, with an IC_50_ of 3.5 μM, whereas the IC_50_ values of falcarinol towards the other cell lines were between 29.2–31.8 μM. This study confirms that falcarinol is one of the most cytotoxic polyacetylenes in apiaceous vegetables.

The roots of cow parsley (*Anthriscus sylvestris*) have been used in Korean folk medicine as an antitussive and diuretic, whereas the leaves are eaten raw, cooked as a potherb or used as a flavoring. A bioassay-guided fractionation of a root extract of the plant using human colon (Colo 205) and human leukemia (K562) cancer cells led to the isolation of several cytotoxic compounds, of which falcarindiol was one of the most cytotoxic with IC_50_ values of 0.819 μg/mL (Colo 205) and 0.577 μg/mL (K562) [147]. Falcarindiol has also been isolated from the fruits by a bioassay-guided fractionation approach using MK-1 cells, and the antiproliferative activity of falcarindiol was tested against MK-1, HeLa and murine melanoma (B16F10) cells and gave the following ED_50_ values 2.8, 55.3 and 18.3 μg/mL, respectively [148]. Falcarindiol is also present in the edible leaves of the plant.

Falcarindiol has also been isolated as a cytotoxic constituent from *Crithmum maritimum* L., commonly known as sea fennel or rock samphire and *Peucedanum japonicum* also known as coastal hog fennel [149,150]. The former has been used in folk medicine for the treatment diuretic, antiscorbutic, digestive, and purgative properties and is consumed as a condiment. Falcarindiol isolated from *C. maritimum* was tested for its cytotoxicity against normal small intestinal epithelial cells (IEC-6) with an IC_50_ value of 20 μM [149]. The leaves of *Peucedanum japonicum* is used in Korean cuisine as a vegetable in various dishes and in Japan it is used as a health food with medicinal properties. The roots of *P. japonicum* is, however, mainly used for medicinal purposes and it has been demonstrated that falcarindiol isolated from the roots inhibit the growth of human leukemia Jurkat T and human promyelocytic leukemia HL-60 cells with an IC_50_ value of 7 μg/mL. In addition, it was demonstrated that falcarindiol inhibited mammalian DNA topoisomerase I at a concentration above 30 μg/mL [150]. DNA topoisomerase is an essential enzyme that control the changes in DNA structure and an inhibition of this enzyme may eventually lead to apoptosis and cell death. Some natural topoisomerase I inhibitors such as the alkaloid camptothecin are used in cancer chemotherapy.

Finally, a cytotoxic panaxydiol-type polyacetylene named cadiyenol (**66**) has been isolated from the aerial parts of Indian pennywort (*Centella asiatica* (L.) Urban), which is used as a culinary vegetable and as a medicinal herb. This compound induced apoptosis (63%) independent of cell cycle regime in mouse lymphoma cells (P388D1) at 28 μM (IC_50_ = 24 μM) within 24 h [151].

#### 2.3.2. Cytotoxic C_17_ and C_18_ Acetylenic Oxylipins from Apiaceae Medicinal Plants

Many plants belonging to the Apiaceae have a long tradition of use in traditional medicine, even though some species are extremely toxic, such as *Bupleurum longiradiatum* Turcz., spotted water-hemlock (*Cicuta maculata* L.), water-hemlock (*C. virosa* L.), and hemlock water-dropwort (*Oenanthe crocata* L.) [9,152]. The former is even not allowed to be used in herbal medicine due to its toxicity. Characteristic for toxic plant members of especially the genera *Bupleurum*, *Cicuta* and *Oenanthe* are the presence of highly neurotoxic C_17_ and C_18_ acetylenic oxylipins, of which some have also been shown to be cytotoxic.

From a dichloromethane extract of the whole plant *B. longiradiatum* 14 structural related acetylenic oxylipins were isolated and tested for their cytotoxicity against the human leukemia cell line HL-60 but only bupleurotoxin (**102**) and its acetate derivative (**103**) [Figure 5] were found to be cytotoxic, with IC_50_ values of 9.4 and 4.9 µM, respectively [153]. The remaining acetylenic oxylipins isolated had IC_50_ values > 10 µM and were considered to be more or less inactive [153] and are therefore not discussed further here. Bupleurotoxin is closely related to oenanthotoxin (**94**) that occurs in *Oenanthe* species such as *O. crocata* [154]. Oenanthotoxin has been tested for its cytotoxicity against six different human cancer cell lines but did only show significant cytotoxicity against an ovarian adenocarcinoma (A2780) cell line with an EC_50_ (half maximal effective concentration) value of 3.8 μM, and moderate cytotoxicity against thyroid carcinoma (8505C) and alveolar basal epithelial adenocarcinoma (A549) cell lines with EC_50_ values around 10 μM [155]. In the same study, synthetic analogues of oenanthotoxin did not result in any improvement of the cytotoxicity compared to oenanthotoxin, and therefore the results are not included. However, a full description of their activity can be found in [155]. Based on the above studies, it appears that the toxic polyacetylenes from *Bupleurum* and *Oenanthe* species are not obvious sources for interesting new cytotoxic acetylenic oxylipins.

The roots and leaves of *Circuta maculata* have been used as an herbal remedy for the treatment of scirrhous mammary cancer and scirrhous tumors [156]. Bioassay-guided fractionation of the active methanol extract of the whole plant using human nasopharyngeal epidermoid carcinoma (KB) cells for the determination of in vitro cytotoxicity resulted in the isolation of cicutoxin (**93**) with an ED_50_ value of 2 μg/mL corresponding to 7.75 μM. Cicutoxin is also a major polyacetylene constituent in the related species *C. virosa* [157]. Testing of various synthetic analogues of cicutoxin (acetates, a monobenzoate, a benzyloxymethyl ether and a keto alcohol) for cytotoxicity resulted in significant less cytotoxicity compared to cicutoxin with ED_50_ (KB) values > 4 μg/mL [156]. Based on these results, it can be concluded that the structural requirements for cytotoxicity of cicutoxin is the presence of the conjugated double and triple bonds as well as the two hydroxyl groups [156]. This conclusion fits well with the conclusion of the structure-activity analysis on the closely related oenanthotoxin and its synthetic analogues. Finally, the antileukemic effect of cicutoxin was also demonstrated in mice [156].

More than 60 species of the genus *Angelica* are medicinal plants and many of these have long been used in ancient traditional medicine, especially in the Far East, such as *Angelica furcijuga*, *A. japonica*, *A. koreana*, and *A. sinensis*, which have been used as remedy against inflammations and viral and bacterial infections. *Angelica sinensis* (Oliv.) Diels is also often used in traditional Chinese medicine for cancer treatment [158]. A bioassay-guided fractionation of the methanol extract of the roots of *A. sinensis* using a MTT-assay against the murine leukemia (L1210) and human leukemia (K562) cell lines resulted in the isolation of several cytotoxic constituents, including the acetylenic oxylipins falcarindiol (**5**) and the related C_18_ acetate (**99**). Both compounds were found to be cytotoxic against L1210 cells with ED_50_ values of 2.87 μM for 5 and 2.60 μM for **99**, whereas they were inactive towards K562 cells showing ED_50_ values above 10 μM [159]. Falcarindiol has also been isolated as a cytotoxic constituent of a chloroform extract of the roots of *A. japonica* A. Gray that showed high inhibitory activity against human gastric adenocarcinoma (MK-1) cell growth [160,161]. Falcarindiol was isolated from the extract by a bioassay-guided approach together with four furanocoumarin ethers of falcarindiol (**12**–**15**), falcarinol (**1**), falcarindiol 8-acetate (**7**) and (9*Z*)-heptadeca-1,9-dien-4,6-diyn-3,8,11-triol (**16**). The cytotoxicity of the isolated polyacetylenes against MK-1 cells showed that the cytotoxicity of the bulky furanocoumarins of falcarindiol **12**–**15** with ED_50_ values of 5.0–16.2 μg/mL were lower compared to falcarinol (ED_50_ = 0.3 μg/mL), falcarindiol (ED_50_ = 3.9 μg/mL), and falcarindiol 8-acetate (ED_50_ = 3.2 μg/mL), and (9*Z*)-heptadeca-1,9-dien-4,6-diyn-3,8,11-triol (ED_50_ = 2.2 μg/mL) [160]. In the same study, the antiproliferative effects of the isolated polyacetylenes were also tested against HeLa and B16F10 cells but were found to be relatively inactive with ED_50_ values > 15 μg/mL for most of the isolated polyacetylenes. For falcarindiol, these test results are in accordance with those obtained for falcarindiol isolated from *Anthriscus sylvestris* described in Section 2.3.1. Other medicinal plants belonging to Apiaceae, where falcarindiol and/or falcarinol have been isolated as the main cytotoxic constituents are *Chaerophyllum hirsutum* L. [162], *Cnidium officinale* Makino [163], *Glehnia littoralis* F. Schmidt *ex* Miq. [140,141,164], *Heracleum moellendorffii* Hance [165], and *Saposhnikovae divaricata* (Turcz.) Schischk. [142,166]. For example in *C. officinale* falcarindiol was found to have antiproliferative effects against MCF-7 human breast cancer cells by induction of a G_0_/G_1_ cell cycle arrest of the cells, with an IC_50_ value of 35.67 μM. In addition, falcarindiol induced apoptosis through strongly increased mRNA expression of Bax and p53, and slightly reduced Bcl-2 mRNA levels in a dose-dependent manner [163]. These results are in accordance with a study of the cytotoxicity of an ethanol extract of the roots/rhizomes of *C. officinale*. The extract was shown to inhibit the proliferation of human colon cancer cells (HT-29) in a dose- and time-dependent manner. The mechanism of action for this antiproliferative effect was demonstrated to be via G_1_ phase arrest leading to apoptosis by increasing the expression of p53, p21, Bax and caspase-3, as well as down regulation of the antiapoptotic factor Bcl-2 [167]. The results indicate that *C. officinale* root extract possesses anticancer properties and that falcarindiol is the cytotoxic agent that can explain its cytotoxic activity.

*Notopterygium incisum* Ting ex H. T. Chang is a traditional Chinese herb used for the treatment of inflammation-related diseases, and extracts of the roots and rhizomes of the plant have been shown to exhibit potent cytotoxicity [168]. Phytochemical investigations of the underground of this plant have led to the isolation of a large number of C_17_ and C_18_ acetylenic oxylipins [25,168]. Fourteen of these acetylenic oxylipins have been evaluated for their cytotoxicity (**1**, **4**, **5**, **7**–**9**, **19**, **55**, **57**, **58**, **62**, **95**, **97**, **104**), including a synthetic (3*S*)-isomer of notopolyenol A (**56**), against the cancer cell lines, MCF-7, non-small-cell lung cancer cell line H1299, and HepG2, using the sulforhodamine B assay and taxol as a positive control [168]. The IC_50_ values of the tested polyacetylenes are presented in Table 2. Although most of the tested polyacetylenes did not show significant cytotoxicity against the tested cell lines, it is interesting to note that the synthetic enantiomer of notopolyenol A (**56**) showed high cytotoxicity from 0.6 μM to 1.6 μM in a dose around 24-fold lower than that of the naturally occurring (3*R*)-isomer (**55**). The latter clearly indicate the importance of (3*S*)-configuration for the cytotoxic effect, which is in accordance with the results from the structure-activity analysis for the cytotoxicity of C_17_ polyacetylenes isolated from *P. quinquefolius* and their synthetic enantiomers as described in Section 2.1.1. In addition, the panaxydiol-type polyacetylenes (**57, 58, 62**), with IC_50_ values of 7.3–24.9 μM, displayed stronger inhibitory effects on the cancer cells than those of most of the related falcarinol-type polyacetylenes (**1, 4, 5, 7–9, 19**) and their reduction products (**95** and **104**) [Table 2]. This suggests that the conjugated system enlarged by the 8*E*-double bond may play a positive role the cytotoxicity of panaxydiol-type polyacetylenes [168].

The roots of *Seseli mairei* Wolff, is used in traditional Chinese medicine for the treatment of inflammation, swelling, rheumatism, pain, and the common cold. From an ethanol extract of the roots of this plant showing significant cytotoxicity (ED_50_ < 20 µg/mL) against KB and murine lymphocytic leukemia (P388 and L1210) tumor cells, a cytotoxic panaxydiol-type polyacetylene seselidiol (**63**) was isolated by bioassay-guided fractionation [169]. The isolated seselidiol was tested against, KB, P388, L1210 and human colon carcinoma (HCT-8) cells and showed significant cytotoxic effects with the following ED_50_ values 1.0, 4.9, 3.0 and 10 μg/mL, respectively. The anticancer drug etoposide was used as a positive control (ED_50_ values of 0.1–2.6 μg/mL). Acetylation of seselidiol to seselidiol 3,10-diacetate (**64**) did not affect the cytotoxicity significantly showing ED_50_ values ranging from 4.0 to 7.8 μg/mL, whereas oxidation to 3,10-dioxo-seselidiol (**65**) reduced the cytotoxicity significantly for all tested cell lines, except HCT-8, showing ED_50_ values > 10 μg/mL [169]. Complete saturation of seselidiol resulted as expected in an inactive compound. The results clearly indicate that the unsaturated bonds, i.e., terminal double bond and the diyn-ene chromophore is important for the cytotoxicity of seselidiol. Furthermore, the secondary hydroxyl groups also seems to play a role in the cytotoxicity of seselidiol. The results are in accordance with the structure-activity relationship studies of acetylenic oxylipins discussed in previous sections of this review.

### 2.4. Cytotoxic C_17_ and C_18_ Acetylenic Oxylipins from Other Plant Families

Although polyacetylenes are only widely distributed in the plant families discussed in the previous sections, a few cytotoxic C_17_ and C_18_ acetylenic oxylipins have been isolated from plants of other terrestrial plant families of which some are related to those occurring in Apiaceae and Araliaceae.

The importance of unsaturated double bonds and, in particular, a terminal double bond for the cytotoxicity of C_17_ acetylenic oxylipins, as described in previous sections, was confirmed in a study on polyacetylenes isolated from the roots of *Swietenia macrophylla* King (Meliaceae), commonly known as mahogany. The investigation of potential cytotoxic compounds from the roots of this plant led to the isolation of several panaxydiol-type polyacetylenes that did not possess any terminal double bond [170]. All compounds isolated were tested for their cytotoxicity against the human hepatocellular carcinoma cell line BEL-7402, human leukemia cell line K562, and human gastric carcinoma cell line SGC-7901. Only the panaxydiol-type polyacetylenes **70** and **71** showed moderate cytotoxicity against the above three human cancer cell lines, with IC_50_ values ranging from 14.3 to 45.4 µM, whereas the other panaxydiol-type polyacetylenes (chemical structures not shown) were inactive (IC_50_ > 50 μM) [170]. From *Toona ciliata* var. *ciliate* M. Roem., another plant species belonging to Meliaceae the cytotoxic panaxydiol-type polyacetylene **69** was isolated from an ethyl acetate extract of the leaves and tested for in vitro inhibitory activities against HL-60, SMMC-7721, A549, SK-BR-3, and PANC-1 human tumor cell lines. However, significant cytotoxicity was only observed against the HL-60 cells with an IC_50_ value of 6.7 μM. Two other related C_18_ acetylenic oxylipins that were also isolated from the plant were inactive [171].

The stems and leaves of *Scurrula atropurpurea* (Blume) Danser (Loranthaceae), a parasitic plant that attacks the tea plant *Thea sinensis* (Theaceae), have been traditionally used for the treatment of cancer in Java (Indonesia) [172]. Three C_18_ acetylenic oxylipins octadeca-8,10-diynoic acid (**107**), (*Z*)-octadec-12-en-8,10-diynoic acid (**108**), and octadeca-8,10,12-triynoic acid (**109**) (Figure 6) were isolated from 70% aqueous acetone extracts of stems and leaves. All compounds showed inhibitory activity against cancer cell invasion (MM1 cells) in vitro in a concentration of 10 μg/mL and **109** even down to 5 μg/mL with 95% inhibitory activity [8,173].

Another polyacetylenic C_18_ acid that has shown interesting cytotoxic activities is minquartynoic acid (**111**) (Figure 6), which was initially isolated from the stem bark of *Minquartia guianensis* Aubl. (Oleaceae), also known as Black Manwood or Huambula, by bioassay-guided fractionation using the lymphocytic leukemia cell line P388. Minquartynoic acid was found to inhibit P388 cells with an ED_50_ value of 0.18 μg/mL [174]. Later, minquartynoic acid was isolated from the air-dried bark of *Coula edulis* Baill., also known as African walnut (Oleaceae) [175], and the twigs of *Ochanostachys amentacea* (Olacaceae) [176,177]. The twigs of *O. amentacea* was also found to contain two other cytotoxic minquartynoic acid derivatives, (*E*)-15,16-dihydrominquartynoic acid (**110**) and 18-hydroxyminquartynoic acid (**112**) (Figure 6). All three compounds were tested against a panel of 10 human tumor cell lines (BC1, Lu1, Col2, KB, KB-V+, KB-V-, LNCaP, SW626, SKNSH, M109) and found to be significantly cytotoxic with ED_50_ values ranging from 0.3 to > 20 μg/mL (**110**), 1.4 to 5.5 μg/mL (**111**), and 2.6 to > 20 μg/mL (**112**) [176]. However, 110 exhibited the most potent activity among the three polyacetylenes against the KB, LNCaP (prostate cancer), and SW626 (ovarian cancer) cell lines [176]. Based on the test results it appears that a terminal methyl group at C-18 augment cytotoxic activity of these C_18_ polyacetylenic acids.

Finally, the two cytotoxic C_18_ polyacetylenic acids, octadeca-9,11,13-triynoic acid (**105**) and octadeca-17-en-9,11,13-triynoic acid (**106**) (Figure 6) were isolated from an organic fraction of a defatted aqueous methanol extract of the stem bark of *Mitrephora glabra* Scheff. (Annonaceae) by bioassay-guided fractionation using a human oral epidermoid carcinoma KB bioassay [178]. The C_18_ polyacetylenic acids **105** and **106** were tested against a panel of cancer cell lines including KB, MCF-7, NCI-H460 (human large cell lung carcinoma) and SF-268 (human astrocytoma) cells with IC_50_ values ranging from 10 to 40 µM. The methyl ester of **105** was also isolated and tested but was completely inactive suggesting that the methyl ester diminishes cytotoxicity [178].

### 2.5. Moieties and Stereochemistry that Are Important for the Cytotoxicity of C_17_ and C_18_ Acetylenic Oxylipins

The cytotoxicity of C_17_ and C_18_ acetylenic oxylipins depends on many factors such as their reactivity towards biological nucleophiles, receptor/ligand binding affinity as well as their rate of metabolization. Based on structure–activity analysis studies on the cytotoxic effects of C_17_ and C_18_ acetylenic oxylipins discussed in Section 2.1, Section 2.2, Section 2.3 and Section 2.4, it appears that the diyne functionality is essential for their cytotoxic effects, which is also an important requirement for their reactivity towards biological nucleophiles such as amino acids in proteins (Figure 1). Larger acetylenic chromophores such as triynes and tetraynes contribute to the electrophilic nature of C_17_ and C_18_ acetylenic oxylipins, which may explain their cytotoxic activity. In addition, the configuration of asymmetric centers is important for the cytotoxicity of C_17_ and C_18_ acetylenic oxylipins and in particular, the asymmetric center at C-3 where (3*S*)-configurated acetylenic oxylipins appears to be more cytotoxic than the (3*R*)-isomers. Synthetic compounds containing a 1,3-butadiynylcarbinol motifs have been identified as leads for cytotoxicity against cancer cell lines such as HCT116, HepG2 and Hela cancer cells, and these studies also confirm the importance of the configuration of the chiral center of the secondary alcohol group, where the (*S*)-isomers are generally more cytotoxic than the corresponding (*R*)-isomers [179,180]. The latter is also in accordance with another study on the cytotoxicity of synthetic chiral acetylenic lipids against various cancer cell lines [181]. However, the configuration at C-3 in most C_17_ and C_18_ oxylipins is not critical for their activity because polyacetylenes with *R*-configuration also exert significant cytotoxicity. A terminal double bond in position 1,2 appear, however, to be critical for their cytotoxicity as reduction of the terminal double bond to the corresponding dihydro-derivatives reduces their cytotoxicity significantly. Acetylation and oxidation of the secondary allylic alcohol also diminishes their cytotoxicity and a terminal allylic secondary alcohol is therefore considered important but not critical for the cytotoxicity of C_17_ and C_18_ acetylenic oxylipins. Oxidation of the double bond at C-9 and C-10 in, for example, falcarinol-type polyacetylenes to an epoxide or dihydroxy-derivative is not important for activity, although epoxides are known to be reactive towards nucleophiles. A rearrangement of the 9*Z*-double bond in falcarinol-type polyacetylenes to an 8*E-*double bond as in panaxydiol-type polyacetylenes appears to enhance the cytotoxicity, although further studies are needed in order to conclude on the importance of the rearrangement from a 9*Z* to an 8*E*-double bond for the cytotoxicity of C_17_ and C_18_ acetylenic oxylipins. Finally, the length of the aliphatic carbon chain appears to be important because C_17_ acetylenic oxylipins seem to be more active than the corresponding C_18_ analogues (see Section 2.1.2) and furthermore, it has been shown that the cytotoxicity drops off dramatically in both natural and synthetic polyacetylenes as the chain length shortens, as described in Section 2.1.1. The moieties, stereochemistry, and other structural requirements that are important for the cytotoxicity of C_17_ and C_18_ acetylenic oxylipins are summarized in Figure 7; thus indicating important leads for the development of new potential acetylenic anticancer drugs.

## 3. In Vitro Anti-Inflammatory Activity of C_17_ and C_18_ Acetylenic Oxylipins

From Section 2, it is clear that many C_17_ and C_18_ acetylenic oxylipins exhibit strong cytotoxic activity towards various types of cancer cells and thus may have a chemopreventive effect. The mechanisms of action for the cytotoxic activity of these acetylenic oxylipins may vary depending on the cancer type, but one of the driving mechanisms in the development and progression of cancer is the formation of proinflammatory cytokines and enzymes, as described in the introduction. Despite the existence of a clear connection between cancer and chronic inflammation, it is striking that only a few investigations in the literature have studied the anti-inflammatory effects of these highly bioactive secondary metabolites. In addition, many of the medicinal plants belonging to the Apiaceae, Araliaceae, and Asteraceae families, where C_17_ and C_18_ acetylenic oxylipins are common, have been used to treat inflammatory diseases, which furthermore indicates that these compounds possess interesting anti-inflammatory activities.

As described in the introduction, the transcription factor NF-κB plays a key role for the inducible expression of genes mediating proinflammatory effects; thus, NF-κB signaling pathway inhibitors are not only potential anti-inflammatory but also anticancer drug candidates [182,183,184]. A large variety of inflammatory stimuli leads to NF-κB activation, including carcinogens, lipopolysaccharide (LPS), nitric oxide (NO) and proinflammatory cytokines, such as IL-6, IL-1β and TNF-α [182,183]. In a study by Metzger et al. [185], it was demonstrated that fractionation of a methanol extract of purple carrots by solid phase chromatography on a Sephadex LH-20 column resulted in a fraction that reduced LPS inflammatory response in macrophage and porcine aortic endothelial cells. The active fraction showed a dose-dependent reduction in NO production and mRNA of the proinflammatory cytokines IL-6, IL-1*β*, and TNF-α as well as NO synthase (iNOS) in macrophage cells. In porcine aortic endothelial cells treated with 6.6 and 13.3 µg/mL of the active fraction, the protein secretions of IL-6 and TNF-α were reduced by 77% and 66%, respectively. The active fraction was shown to consist of the polyacetylenes falcarinol, falcarindiol and falcarindiol 3-acetate, which were shown in the study to reduce NO production in macrophage cells by as much as 65% without cytotoxicity. These results clearly demonstrate that the anti-inflammatory compounds in carrots are falcarinol-type polyacetylenes. Falcarinol and falcarindiol isolated from a methanolic extract of the roots of *Angelica furcijuga* Kitagawa was also found to inhibit NO production induced by LPS in cultured mouse peritoneal macrophages with IC_50_ values of 4.8 and 4.4 μM, respectively [186,187]. In addition, falcarindiol inhibited TNF-α with an IC_50_ value of 48 μM [187]. Falcarindiol has also been shown to dose-dependently (1, 5 and 10 μM) reduce the inducible iNOS-mediated NO production without cytotoxic effects on LPS-activated BV-2 microglia cells and microglia [188], and to inhibit NO release from LPS-stimulated RAW 264.7 macrophage cells, with an IC_50_ value of 4.31 μM [163]. In a study of falcarindiol isolated from the stem of *Oplopanax elatus* it was demonstrated that falcarindiol inhibited the formation of NO in LPS-induced RAW 264.7 macrophage cells with an IC_50_ value of 1.28 μM. Other anti-inflammatory polyacetylenes isolated from *O. elatus* that showed to inhibit NO formation were oploxyne A (**39**) and oplopandiol (**18**) with IC_50_ values of 1.98 and 2.72 μM, respectively. All three polyacetylenes also showed inhibition of the major PG, PGE_2_, in LPS-induced macrophages with IC_50_ values in the range of 1.54 to 3.08 μM, which indicate that they inhibit COX enzymes [109]. For falcarindiol, this is in accordance with other studies as described in next paragraph. In addition, the panaxydiol-type polyacetylene cadiyenol (**66**) has been shown to inhibit NO production in a dose-dependent manner in mouse RAW 264.7 macrophage cells, reducing NO production by 70% at 24 μM [151].

Falcarindiol appears to be an effective inhibitor of COX-2 in vitro, as demonstrated in colon cancer and colon epithelial cells [141,189] and in LPS-stimulated RAW 264.7 macrophage cells [163], as well as a an effective COX-1 inhibitor [190,191,192], where an IC_50_ value of 0.3 μM has been reported [190]. Investigations on the COX inhibitory effects of other polyacetylenes are limited; however, it has been shown that falcarinol and panaxydiol significantly inhibit the mRNA expression of COX-2 in colon cancer cells as well as iNOS, but their activity was less compared to falcarindiol [141]. Inhibitory activity of COX-1 has also been demonstrated for falcarinol, falcarindiol 8-acetate (**7**) and dehydrofalcarindiol (**73**), and again the COX inhibitory activity of these polyacetylenes was considerably less compared to falcarindiol [191,193]. In addition, falcarinol and falcarindiol have shown to be strong inhibitors of LOXs (5-, 12- and 15-LOX), which, like COX enzymes, are involved in tumor-progression processes [193,194,195]. In a study by Alanko et al. [193], the IC_50_ values for inhibition of 5-LOX, 12-LOX of leukocyte-type and platelet-type and 15-LOX for falcarinol were 2, 1, 67 and 4 µM, respectively. For falcarindiol the IC_50_ values were 7, 48, > 100 and 18 µM, respectively, which indicates that falcarinol is a more efficient inhibitor of LOXs than falcarindiol. Finally, falcarinol and falcarindiol have also been shown to inhibit the formation of thromboxanes and thus have antiplatelet properties [196,197,198].

Investigations of the anti-inflammatory activity of C_17_ and C_18_ acetylenic oxylipins, indicates that this bioactivity may play an important role in their anticancer effects, at least for some cancers such as CRC where inflammation plays a central role in the development and progression of this disease. This has recently been demonstrated for falcarinol and falcarindiol in a rat model of CRC as described in Section 4.1. In addition, falcarinol has been shown to reduce induced acute intestinal and systemic inflammation in a mice model by exerting anti-inflammatory activity through activation of the Keap1-Nrf2 pathway, as described in more detail in Section 4. However, it is clear that information about the potential anti-inflammatory effects of C_17_ and C_18_ acetylenic oxylipins is lacking in the literature, with a few exceptions. Therefore, it is suggested that future investigations on the anticancer effect of these bioactive acetylenic oxylipins should focus more on their direct anti-inflammatory activity as well as the anti-inflammatory effect related to the activation of the Keap1-Nrf2 pathway. This is likely to give new important knowledge and insight into the mechanisms of action of the chemopreventive effects of C_17_ and C_18_ acetylenic oxylipins.

## 4. In Vivo Anticancer Activity of C_17_ and C_18_ Acetylenic Oxylipins

### 4.1. In Vivo Studies of the Chemopreventive Effect of Falcarinol and Falcarindiol in a Rat Model for CRC

Studies on the in vivo anticancer effect of acetylenic oxylipins are few and the most detailed in vivo studies concerns the investigation of the cancer-preventive effects of carrots and their main polyacetylenic constituents, falcarinol and falcarindiol, in relation to the development of CRC. As mentioned in Section 2.3.1 there is evidence from meta-analysis and cohort studies that carrots have cancer-preventive effects against several types of cancers, including CRC. There is strong evidence for the existence of a link between CRC development and the expression of COX enzymes, and in particular COX-2 [37]. COX-2 overexpression but not COX-1 has been detected in 50% of colorectal adenomas and in up to 85% of colorectal carcinomas, and it is correlated with poor prognosis, and therefore COX-2 is considered as one of the most important drug targets for the prevention of CRC [39]. This is consistent with the fact that CRC development often is linked with chronic inflammation resulting in a microenvironment conducive for the development of early neoplastic lesions due to relative high concentrations of PGs [199]. Nonsteroidal anti-inflammatory drugs (NSAIDs) such as aspirin are known COX inhibitors, and they have been widely used for cancer prophylaxis [200]. However, several adverse effects have been associated with the long-term usage of NSAIDs such as gastrointestinal bleeding, asthma, hepatic, renal and cardiovascular toxicity, primarily due to their irreversible inhibition of COX enzymes and in particular COX-1; hence, their clinical application has been limited [201]. Consequently, dietary secondary metabolites such as polyacetylenes that are known for their cytotoxic and anti-inflammatory activities have therefore been considered as promising chemopreventive candidates because they inherently show low toxicity and a promising safety profile with no known severe side effects. Falcarinol has so far only shown unwanted toxic effects when delivered in high doses upon injection (100 mg/kg) to rodents, where it causes neurotoxic symptoms [202], whereas falcarindiol does not seem to have any toxic effect [203]. Unwanted toxic effects of falcarinol and falcarindiol in humans by even a relatively high intake of carrots and/or processed carrot products or other apiaceous vegetables is therefore not expected and this is the background for the investigation of the cancer-preventive effects of these major dietary polyacetylenes in CRC rat models [52,53,204].

So far, three preclinical trials have been performed on falcarinol and falcarindiol isolated from carrots in azoxymethane (AOM)-induced rats, which is an accepted in vivo CRC model [52,53,204,205]. AOM is a potent carcinogen, which is an efficient inducer of aberrant-crypt foci (ACF) and other precancerous lesions as well as CRC in rats [53,205]. Only a small number of early neoplastic ACF lesions will develop to neoplastic polyp lesions and only a small number of these lesions will have the potential to develop into adenomas and cancers [206,207]. All three clinical trials demonstrated chemopreventive effects on CRC [52,53,204]. In the most recent clinical trial the dose-dependent antineoplastic effect of purified falcarinol and falcarindiol in AOM-induced rats was investigated as well as their possible mechanisms of action. Groups of 20 rats received a standard rat diet (SRD) supplemented with 0.16, 0.48, 1.4, 7 or 35 µg falcarinol and falcarindiol/g feed in the ratio 1:1, respectively, and 20 rats were controls receiving only SRD [52]. The ratio 1:1 of falcarinol and falcarindiol in the rat feed was based on the synergistic cytotoxic effects of these polyacetylenes described in Section 2.3.1.

Analysis of aberrant crypt foci (ACF) in the above study showed that the average number of small ACF (< seven crypts) and large ACF (> seven crypts) decreased with increasing dose of falcarinol and falcarindiol and that this inhibitory effect on early neoplastic formation was dose-dependent (Table 3). A dose-dependent effect on the total number of macroscopic neoplasms was also observed (Table 3). Not surprisingly, the largest antineoplastic effect was observed at the highest doses of falcarinol and falcarindiol in the feed. Interestingly, adenomas were, in general, smaller in the falcarinol- and falcarindiol-treated rats, especially at the highest doses compared to the control group [52]. This clinical trial clearly demonstrated a dose-dependent chemopreventive effect of falcarinol and falcarindiol on the formation of neoplastic lesions in the colon of rats. In addition, the study also indicated that these dietary polyacetylenic oxylipins might exert a growth inhibition of neoplastic lesions, as they not only reduce the number of macroscopic polyp neoplasms but also their size. The latter suggests that falcarinol and falcarindiol exert a cytotoxic and thus a kind of chemotherapeutic effect. The dose for an optimal antineoplastic effect was found to be from 7 to 35 µg of falcarinol and falcarindiol/g feed in the rat model of CRC. However, effects are observed at even lower concentrations (Table 3) and it appears that a cancer-preventive dose of polyacetylenes in humans, based on this preclinical trial, can be achieved with a daily intake above 30 g raw carrots [52,130], which is in accordance with the results of the cohort study mentioned in Section 2.3.1.

Gene expression studies performed by real-time quantitative PCR on selected cancer biomarkers in neoplastic tissue from the rat model of CRC revealed that falcarinol and falcarindiol downregulated NF-κβ and its downstream proinflammatory markers TNFα, IL-6, and COX-2, whereas biomarkers such as COX-1 and IL-1β were not significantly affected [52]. Furthermore, the gene expression study showed a downregulation of PPARγ. The downregulation of COX-2 in adenomas from rats receiving SRD supplemented with falcarinol and falcarindiol compared to adenomas of the control group was furthermore confirmed by immunohistochemical analysis [52]. A significant downregulation of COX-2 in rats receiving SRD supplemented with falcarinol and falcarindiol implies the involvement of COX-2 in CRC development and, thereby, clearly indicates that COX-2 is an important target for the chemopreventive effects of these polyacetylenes. Based on the results of this preclinical trial, it appears that falcarinol and falcarindiol in combination act as selective COX-2 inhibitors, which is highly interesting in relation to the development of anti-inflammatory and anticancer drugs with no severe side effects.

The significant downregulation of NF-κB in tumor tissue in rats receiving SRD supplemented with falcarinol and falcarindiol compared to the control group confirms that inflammation plays a key role in early neoplastic formation and that the chemopreventive effect of falcarinol and falcarindiol in the colon is linked to their anti-inflammatory activity [52]. In addition, the downregulation of the proinflammatory cytokines TNFα and IL-6 in tumor tissue in rats receiving SRD supplemented with falcarinol and falcarindiol is further evidence for the anti-inflammatory action and chemopreventive effects of these dietary polyacetylenes. An upregulation of TNFα is linked to increased leukocyte infiltration and tumor formation and is usually detected in colorectal neoplasms, and in animal models of CRC [208,209] and IL-6 is highly upregulated in many cancers and is considered as one of the most important proinflammatory cytokines during carcinogenesis and metastasis [48].

The nuclear receptor PPARγ controls the expression of a large number of regulatory genes in inflammation and cell proliferation. In the colon, PPARγ plays a key role in the control of intestinal inflammation such as ulcerative colitis [210]. Furthermore, PPARγ has been found to have antineoplastic properties being able to induce apoptosis and differentiation of colon cancer cells both in vivo and in vitro [211]. The regulation of PPARγ expression in the colon is unresolved but PPARγ expression may be upregulated by intestinal-microbial interactions involving LPS of Gram-negative bacteria and/or ligands of PPARγ [52,211]. A number of acetylenic C_17_ and C_18_ oxylipins, including falcarinol and falcarindiol have been shown to be ligands of PPARγ and to activate this nuclear receptor [23,24,25,212]. This seems to explain their antidiabetic properties and may contribute to their anti-inflammatory and anticancer activities; thus, an increased expression of PPARγ in epithelial cells would have been expected in rats receiving falcarinol and falcarindiol in the diet. The significant downregulation of PPARγ in rats receiving falcarinol and falcarindiol in the diet compared to the control group indicates, however, that PPARγ may not be an important target for the antineoplastic effect in the colon of these polyacetylenic oxylipins. This may be explained by recent research that has shown that falcarinol and falcarindiol alter the gut microbiota composition in the AOM-induced rat model [213], resulting in a downregulation of the expression of PPARγ in the tumor tissue observed in the above preclinical trial instead of an expected upregulation [52].

The anti-inflammatory activities observed in the rat model for CRC may be linked to their ability to activate the Keap1-Nrf2 pathway as recently demonstrated in CB57BL/6 mice. In this study pretreatment of 5 mg falcarinol/kg twice a day for one week against LPS-induced acute intestinal and systemic inflammation was shown to reduce the magnitude of intestinal proinflammatory gene expression (IL-6, TNFα, INFγ, STAT3, and IL-10) [26]. Furthermore, falcarinol was shown to upregulate the cytoprotective enzyme HO-1 in both the intestine and in the liver and to reduce basal lipid peroxidation in the mesentery. Finally, falcarinol was shown to protect against LPS-induced reduction in intestinal barrier integrity as well as reduced inflammatory cell infiltration [26]. Hence, the results from this study to some extent support the results of the anti-inflammatory mechanisms of action for the cancer-preventive effects of falcarinol and falcarindiol observed in the CRC rat model.

Collectively, the results of the above preclinical trial demonstrate that falcarinol and falcarindiol in combination have a dose-dependent chemopreventive effect on neoplastic lesions in a rat model of CRC, and that this effect is most likely due to inhibition of downstream proinflammatory markers in the NF-κB signaling pathway. In particular, COX-2 seems to be an important target for the anti-inflammatory and antineoplastic effect of falcarinol and falcarindiol in relation to the development of CRC.

### 4.2. Studies of the Anticancer Effect of C_17_ and C_18_ Acetylenic Oxylipins in Other In Vivo Models

As described in in Section 2.1.2 (3*S*,8*S*)-falcarindiol (**10**) and oplopantriol A (**97**) isolated from *Oplopanax horridus* has been shown to be cytotoxic to several types of cancer cell lines. The in vivo anticancer effects of these polyacetylenes have been evaluated in a xenograft tumor model using the human CRC cell line HCT-116 [102,106,107]. The in vivo antitumor potential of falcarindiol was evaluated in two studies by inoculating firefly luciferase-tagged HCT-116 cells into the flanks of athymic nude mice [102,107]. Falcarindiol in a concentration of 15 mg/kg was administered intraperitoneally every day starting from day 1. Quantitative analysis of imaging data revealed that falcarindiol significantly inhibited xenograft tumor growth in week 2 (*P* < 0.05) and much more significant in weeks 3 and 4 (*P* < 0.01) after falcarindiol administration compared to vehicle administration (control). In addition, the cytotoxicity of falcarindiol was evaluated on the normal rat small intestine epithelial cell line (IEC-6) but did not show any cytotoxicity even at a concentration of 20 μM, whereas, at 10 μM, the cell growth of HCT-116 cells was almost completely inhibited [102]. Consequently, falcarindiol showed significant antitumor activity in the HCT-116 xenograft tumor model and was relatively safe to normal intestinal cells. The same HCT-116 xenograft tumor model has also been used to evaluate the in vivo anticancer effect of oplopantriol A also in a concentration of 15 mg/kg, where quantitative analysis of imaging data revealed a significant xenograft tumor growth starting from week 3 (*P* < 0.05) and being much more significant at weeks 4 and 5 (*P* < 0.01) [106]. The safety profile of oplopantriol A was not evaluated in this study.

The cytotoxic polyacetylenes isolated from *Dendropanax arboreus* described in Section 2.1.2 have also been investigated for their in vivo activity in a murine LOX melanoma xenograft model [112]. Tumor cells were implanted in the peritoneal cavity of mice in order to provide maximal sensitivity for the preliminary in vivo evaluations, and (3*S*)-falcarinol (**3**), dehydrofalcarinol (**72**) and dehydrofalcarindiol (**73**), formulated in sesame oil, were injected intraperitoneal. All tested polyacetylenes showed modest activity but still some potential for in vivo antitumor activity with dehydrofalcarinol having the greatest activity in the employed LOX melanoma model showing the best effect at a concentration at 3 mg/kg.

In another study, orally administered (3*R*)-falcarinol (**1**) was shown to significantly suppress the initiation and growth of lung cancer in mice models in concentrations of 50 mg/kg, without detectable toxicity [214]. Furthermore, it was demonstrated that falcarinol suppressed carcinogenesis both in vitro and in vivo by disrupting heat shock protein 90 (Hsp90) function and viability of both non-cancer stem-like and cancer stem-like cells of non-small-cell lung cancer by inducing apoptosis without increasing Hsp70 expression. Induction of apoptosis by falcarinol was demonstrated in cancer stem like cells in nM concentrations [214]. Hsp90 is known to promote cancer cell survival and acquisition of anticancer drug resistance by controlling the conformational maturation and stability of numerous carcinogenic proteins, and overexpression of this protein has been observed in many human cancers and is associated with poor prognosis [215,216,217]. Hsp70 is the most ubiquitous stress-inducible chaperone and accumulates in the cells in response to a wide variety of insults including anticancer chemotherapy, and is known to inhibit key effectors of the apoptotic and autophagy machineries. In tumors and cancer cells, the expression of Hsp70 is very high, and Hsp70 may participate in carcinogenesis and in resistance to chemotherapy [218]. The mechanism by which falcarinol disrupted the function of Hsp90 and did not overexpress Hsp70 was shown to occur by binding to the *N*-terminal and *C*-terminal ATP binding pockets of the protein, resulting in destabilization of its client proteins, as shown both in vitro and by molecular docking studies [214]. This in vivo and in vitro study demonstrates that falcarinol is a potential inhibitor of Hsp90 with limited toxicities and interestingly give new insight into the possible mechanisms of action for the anticancer effect of falcarinol and related acetylenic oxylipins.

Panaxydol (**20**), which has shown to induce cell cycle arrest and apoptosis in cancer, as described in Section 2.1.1, has also been investigated for its antitumor effect in vivo in mice tumor models [76]. Syngeneic BALB/c mice were inoculated in the flank with mouse renal cell carcinoma Renca cells one week before administration of 100 or 200 mg/kg panaxydol therapy by i.p. injection every two days for three weeks, which resulted in a dose-dependent reduction of tumor size. Inhibition of in vivo tumor growth was also demonstrated with a human prostate cancer (PC3) xenograft where PC3 cells were inoculated in BALB/c nude mice one week before the initiation of three-week panaxydol therapy (50 or 100 mg/kg every 2 days). This resulted in inhibited growth of the PC3 xenograft dose-dependently, with complete suppression at 100 mg/kg [76]. The results of these in vivo studies, in combination with its in vitro cytotoxicity, suggest that panaxydol is a promising anticancer candidate or lead compound for the development of efficient anticancer drugs.

Finally, in vivo antitumor efficacy of (3*R*,9*R*,10*R*)-panaxytriol (**36**) isolated from *P. ginseng* has been shown to significantly delay tumor growth in C57BL/6 mice (*P* < 0.01) transplanted with B 16 melanomas when administered intramuscularly with panaxytriol in a concentration of 40 mg/kg [80,219]. Furthermore, the in vivo antitumor efficacy of panaxytriol has been evaluated in human breast cancer xenograft (MX-1) mice models, where the mice where treated with panaxytriol at various dosages through intravenous infusion [220]. Mice treated with 30 mg/kg of panaxytriol exhibited some suppression of tumor growth, but no appreciable shrinkage in tumor mass was observed; thus no chemotherapeutic effect. At elevated dosage levels, i.e., at 50 mg/kg, improved inhibitory effects were observed. No changes in body weight were observed upon treatment, which clearly indicate that the antitumor activity achieved with panaxytriol is apparently not accompanied by any toxic side effects, even at high dosage levels.

## 5. Conclusions

The present review has demonstrated that C_17_ and C_18_ acetylenic oxylipins constitute an interesting class of cytotoxic natural products that may be used in the prevention and treatment of cancers and as lead compounds for the development of anticancer drugs. The mechanisms of actions of these acetylenic oxylipins are the same as for many anticancer drugs, such as inducing ER stress, cell cycle arrest, apoptosis, and/or reduce inflammation, but, interestingly, some of the cytotoxic acetylenic C_17_ oxylipins that have been tested in preclinical trials and demonstrated an anticancer effect do not show any significant toxic side effects. However, only a few cytotoxic polyacetylenes have been tested in preclinical trials, and, so far, none of them has been tested in clinical trials, although there is evidence for a likely cancer-preventive effect of dietary polyacetylenes from meta-analysis and cohort studies. Polyacetylenes most often do not occur in large amounts in plants and furthermore they are unstable, being sensitive to heat, light and oxidation [3,7], and, therefore, it is a challenge to isolate relatively large amounts of these compounds from plants for preclinical trials. This may also be one of the explanations for the relatively few preclinical trials performed on these compounds. Still, the present review may encourage more preclinical trials and initiate clinical trials in particular on the dietary polyacetylenes in order to obtain more evidence for their anticancer effect as well as to obtain more information on their pharmacokinetics and mechanisms of action, including their potential targets in cells.

## Figures and Tables

**Figure 1 molecules-25-02568-f001:**
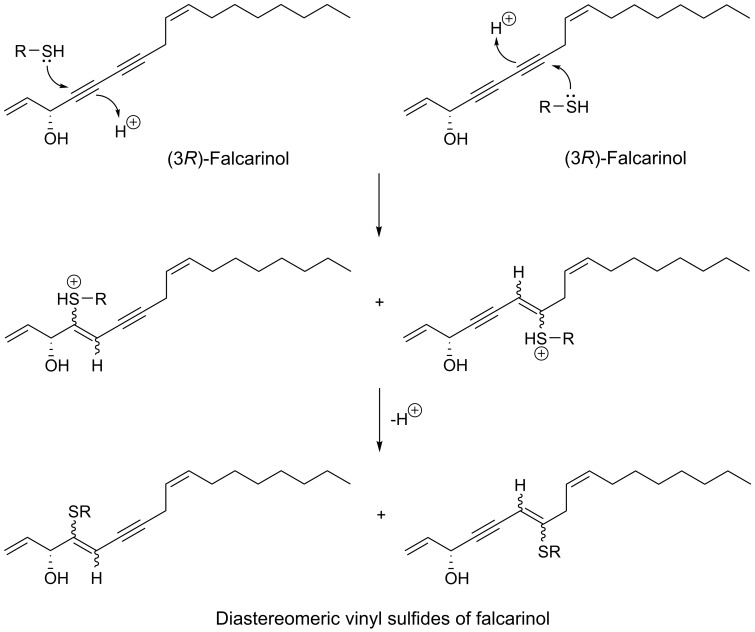
Reaction of (3*R*)-falcarinol with thiols in proteins or peptides (R–SH) leading to a mixture of diastereomeric vinyl sulfides of falcarinol. The covalent binding of electrophilic C_17_ and C_18_ acetylenic oxylipins to nucleophilic biomolecules such as proteins/peptides explains to some extent the cytotoxicity and anti-inflammatory activity of these secondary metabolites.

**Figure 2 molecules-25-02568-f002:**
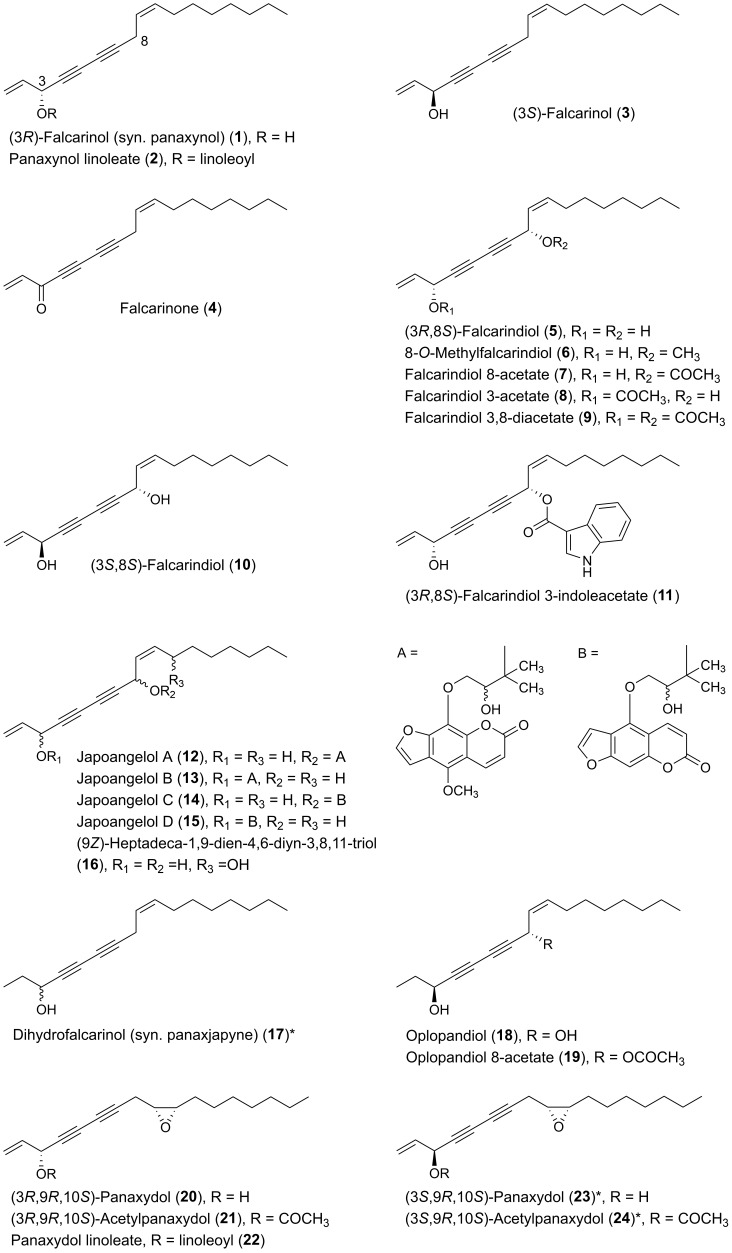
Cytotoxic C_17_ acetylenic oxylipins (**1**–**56**) of the falcarinol-type. An asterisk (*) after a compound number indicates a cytotoxic falcarinol-type polyacetylene of synthetic origin that is not naturally occurring, but which may prove to be present in plants.

**Figure 3 molecules-25-02568-f003:**
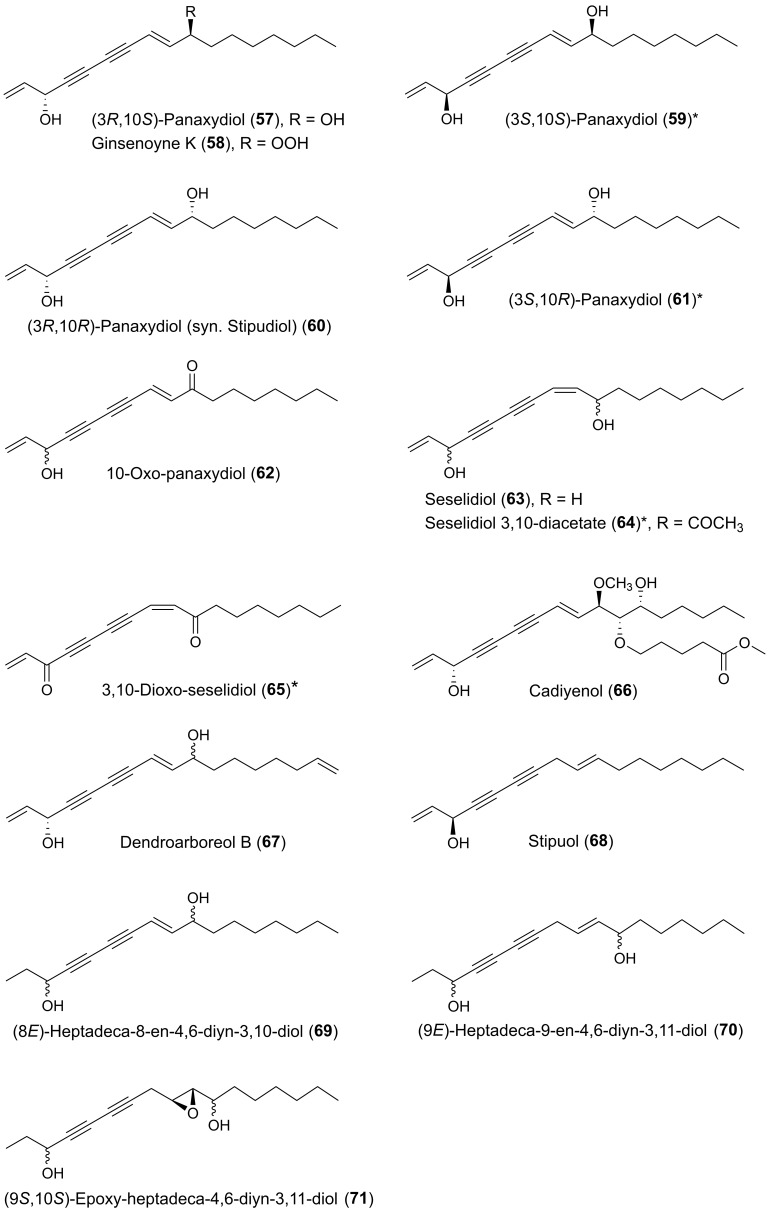
Cytotoxic C_17_ acetylenic oxylipins (**57**–**71**) of the panaxydiol-type. An asterisk (*) after a compound number, indicates a cytotoxic panaxydiol-type polyacetylene of synthetic origin that is not naturally occurring, but which may prove to be present in plants.

**Figure 4 molecules-25-02568-f004:**
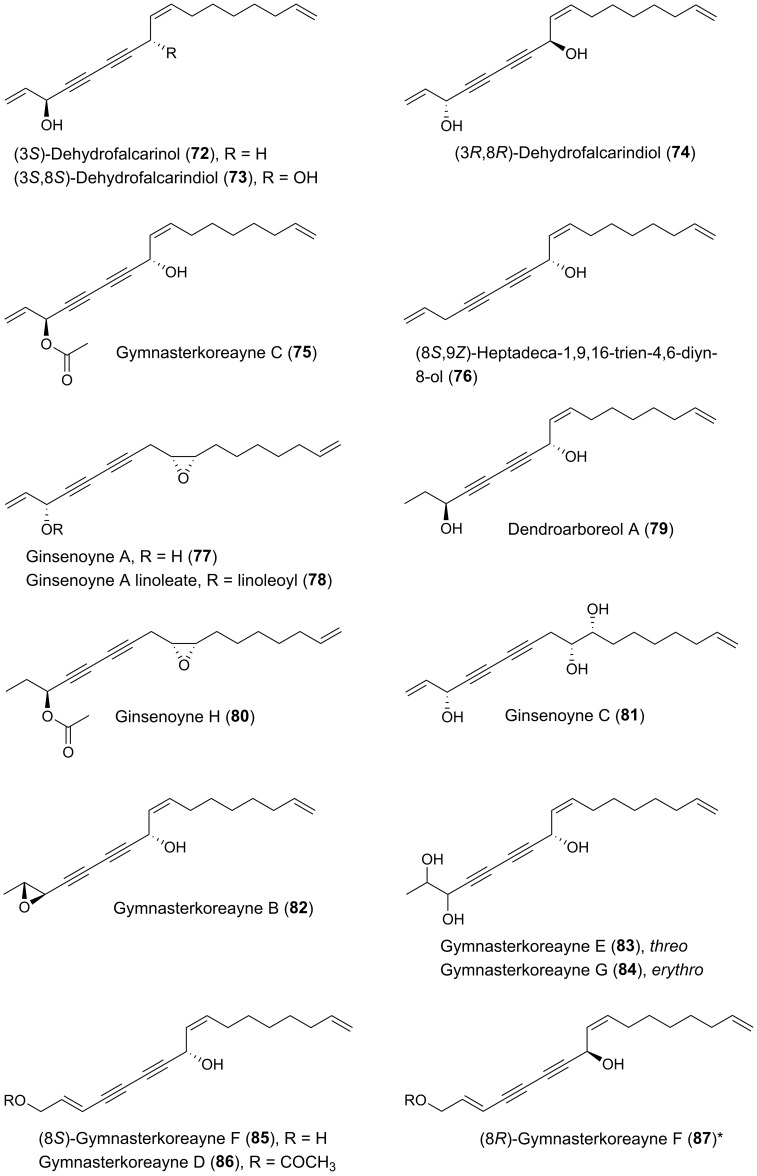
Cytotoxic C_17_ acetylenic oxylipins (**72**–**92**) of the dehydrofalcarindiol-type. An asterisk (*) after a compound number, indicates a cytotoxic dehydrofalcarindiol-type polyacetylene of synthetic origin that is not naturally occurring, but which may prove to be present in plants.

**Figure 5 molecules-25-02568-f005:**
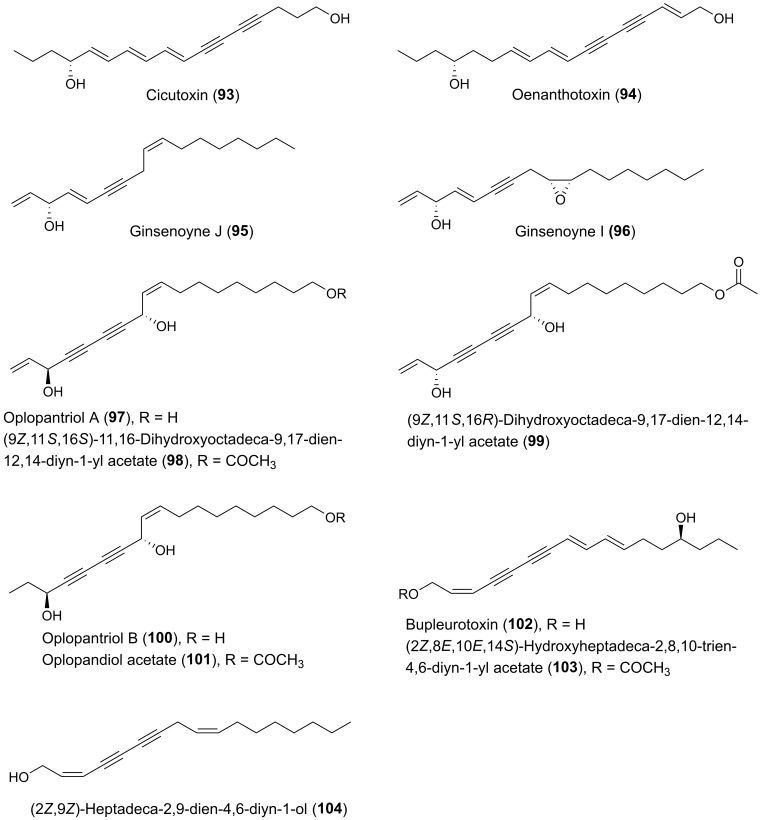
Miscellaneous cytotoxic C_17_ and C_18_ acetylenic oxylipins (**93**–**104**) isolated from medicinal plants belonging to the Araliaceae or Apiaceae families.

**Figure 6 molecules-25-02568-f006:**
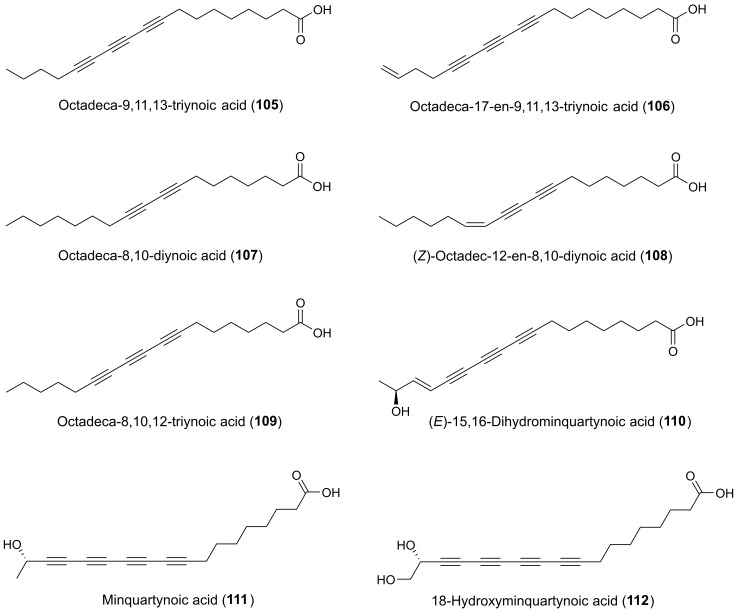
Cytotoxic C_18_ polyacetylenic acid isolated from *Mitrephora glabra* (Annonaceae) (**105**, **106**), *Scurrula atropurpurea* (Loranthaceae) (**107**–**109**) and plants belonging to the Oleaceae family (**110**–**112**).

**Figure 7 molecules-25-02568-f007:**
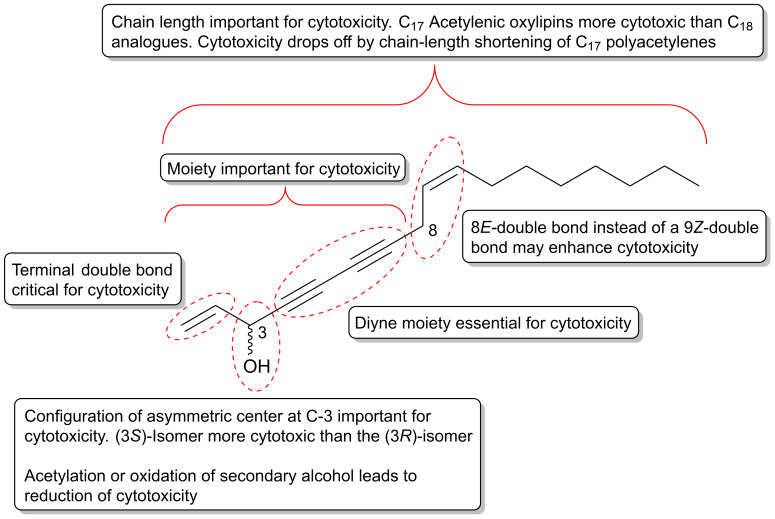
Illustration of moieties, stereochemistry and other structural requirements that are important for the cytotoxicity of C_17_ and C_18_ acetylenic oxylipins (falcarinol as an example) isolated from Apiaceae, Araliaceae, and Asteraceae families based on structural-activity relationship studies.

**Table 1 molecules-25-02568-t001:** Effects of different ratios of falcarinol (**1**) and falcarindiol (**5**) on the proliferation of Caco-2 cells in culture medium containing 0.625% fetal calf serum [136] ^a^.

Falcarindiol (μg/mL)	Falcarinol (μg/mL)			
	0	1	5	10
0	1.0	1.01 ± 0.25	0.36 ± 0.25	0.18 ± 0.01
1	1.35 ± 0.36	0.71 ± 0.08 ^b^	0.58 ± 0.04	0.22 ± 0.08
5	1.03 ± 0.32	0.53 ± 0.06 ^b^	nd ^c^	nd
10	0.52 ± 0.17	0.17 ± 0.03 ^b^	nd	nd

^a^ Mean ± standard deviation is shown for each combination of falcarinol and falcarindiol. Data are presented relative to cell proliferation obtained in medium without polyacetylenes for two individual experiments. ^b^ Significant synergistic inhibitory effect on cell proliferation compared to single-compound assay (*P* < 0.01). ^c^ nd = Not determined.

**Table 2 molecules-25-02568-t002:** Cytotoxic effects of acetylenic oxylipins (**1**, **4**, **5**, **7**–**9**, **19**, **55**, **57**, **58**, **62**, **95**, **97**, **104**) isolated from *Notopterygium incisum* as well as the synthetic (3*S*)-isomer of notopolyenol A (**56**) and the positive control taxol against the cancer cell lines MCF-7, H1299 and HepG2 [168].

Acetylenic Oxylipin	IC_50_ (μM)		
	MCF-7	H1299	HepG2
**1**	43.1 ± 0.1 ^a^	30.8 ± 0.1	45.2 ± 0.2
**4**	> 100	> 100	> 100
**5**	29.4 ± 1.0	22.1 ± 0.9	23.6 ± 2.0
**7**	19.0 ± 0.9	16.4 ± 0.7	15.9 ± 0.7
**8**	29.6 ± 1.9	21.3 ± 1.9	11.7 ± 1.2
**9**	67.8 ± 2.3	37.6 ± 1.3	22.7 ± 0.2
**19**	45.6 ± 1.5	14.6 ± 0.8	20.8 ± 1.2
**55**	31.7 ± 1.3	24.9 ± 0.9	35.3 ± 0.5
**56**	1.3 ± 0.6	0.6 ± 0.2	1.4 ± 0.7
**57**	13.5 ± 1.9	12.8 ± 0.9	24.9 ± 0.6
**58**	7.3 ± 0.4	10.7 ± 0.8	19.2 ± 2.2
**62**	15.1 ± 1.9	12.1 ± 0.9	23.6 ± 2.0
**95**	85.7 ± 0.4	31.9 ± 0.2	54.2 ± 1.6
**97**	> 100	> 100	29.7 ± 2.7
**104**	66.7 ± 1.2	36.0 ± 1.6	47.6 ± 1.9
Taxol	0.0022 ± 0.0003	0.0018 ± 0.0008	0.0020 ± 0.0007

^a^ Mean ± standard deviation.

**Table 3 molecules-25-02568-t003:** Small aberrant crypt foci (ACF) (< seven crypts) and large ACF (> seven crypts) and the total number of macroscopic polyp neoplasms (tumors > 1 mm) in six groups of 20 azoxymethane-induced rats receiving a standard rat diet (SRD) or a SRD supplemented with different doses of falcarinol (**1**) and falcarindiol (**5**) (ratio 1:1). Data are shown as mean ± standard deviation [52].

Size of Neoplasms	µg Falcarindiol and Falcarinol (1:1)/g Feed					
	0 (*n* = 20)	0.16 (*n* = 20)	0.48 (*n* = 20)	1.4 (*n* = 20)	7 (*n* = 20)	35 (*n* = 20)
Mean ACF < 7 crypts ^a^	205 ± 36	207 ± 28	180 ± 29	171 ± 26	150 ± 31	145 ± 19
Mean ACF > 7 crypts ^b^	nd ^c^	14 ± 3.7	12 ± 4.1	10 ± 3.7	nd	8 ± 3.5
Total number of macroscopic polyp neoplasms ^d^	21	18	19	13	12	7

^a^ Significant dose-response effect according to linear regression analysis (*R*^2^ = 0.3742, *P* < 0.001). ^b^ Significant dose-response effect according to linear regression analysis (*R*^2^ = 0.2451, *P* < 0.001). ^c^ nd = no data are available. ^d^ Dose-response relationship determined by Poisson regression with the number of macroscopic lesions decreasing by 0.167 for each log-fold increase in dose (*P* = 0.007).

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
