# Peer review of "Bioactive C17 and C18 Acetylenic Oxylipins from Terrestrial Plants as Potential Lead Compounds for Anticancer Drug Development"

_molecules, 2020, doi:10.3390/molecules25112568_

Round 1

Reviewer 1 Report

The Review-type article from Lars O. Christensen “Bioactive C17 and C18 Acetylenic Oxylipins from Terrestrial Plants as Potential Lead Compounds for Anticancer Drug Development” is a detailed work which covers most of the acetylenic oxylipins and updates the activities both in vitro and in vivo of them. It focused on the chemopreventive activity of the compounds as well as the anti-inflammatory and the direct anticancer benefits. The work is finely written with a clear structure starting with the in vitro activities of oxylipins splitted because of plant origin to follow with the anti-inflammatory assessment and the in vivo anticancer evaluations. Overall, the review includes all the key references from the field covering a long period of time and seek to discuss them in a comprehensive manner. As well the references totally agree with the text, are correctly positioned and are a large number of works (> 220 references). It is evident that the work is an attractive piece of work to publish at Molecules and will be of valuable interest to researchers involved in drug discovery, anticancer agents, anti-inflammatory agents and oxylipins chemistry.

I would just recommend some minor points to increase the quality of the review:

(1) As a general opinion, it is difficult to follow the explanation and take a look to the chemical structures since the compound labelling is in figures that do not follow the document. It will be better to make more figures in a sequential-line manner so that the reader can direct easily to the compounds that the text refers.

(2) The author does not explain in detail the biological molecular targets of these compounds, only a few sentences about ABCG2 and ALDHs. It would benefit for progressing in the field by determining the potential targets in cells and/or a more descriptive mechanism of action. Does the author not cover this because of the lack of these topics? If they exist, the author should explain in much more detail at the introduction.

(3) At some points, the text points out “data no shown” although it will be convenient to change for a statement “see reference X for a fully description of derivatives of X which are not included because of poor activity” or a legend or something similar. The first time looks weird to the reader although the following ones it is clear why is indicated this.

(4) In the figure 2 for the Japoangelol derivatives, from a chemical point of view it makes no sense the drawing and it does not specify with is the atom of A/B conjugated to the oxylipin scaffold. It shall be correctly drawn and place the atom bond of the aromatic moieties with a zipped line or similar.

Some spelling errors:

Line 134, “indicates” by “indicate”

Line 277, shall be written cytotoxicity

Line 370 “appear” instead “appears”

Line 398 “ntis” acronym of ?

Line 466, shall be written pro-apoptosis

Line 591 “resonance” and “carbocation”

Line 729 “indicate”

Line 768 “seems”

Author Response

I would like to thank reviewer 1 for some good comments to the manuscript, which had helped me to improve the quality of the manuscript/review. Below is my response to the comments of this reviewer:

Comment 1:

“As a general opinion, it is difficult to follow the explanation and take a look to the chemical structures since the compound labelling is in figures that do not follow the document. It will be better to make more figures in a sequential-line manner so that the reader can direct easily to the compounds that the text refers.”

Response to comment 1:

The chemical structures of the compounds are arranged in accordance with their chemical structures, i.e., falcarinol type polyacetylenes, panaxydiol type polyacetylenes etc. Although this mean that the compound labelling in the figures do not exactly follow the text in the document, it creates some coherence in the labelling. In my preparation of the manuscript I tried in fact to include more figures in a sequential-line but it creates other problems because then there is no structure in the numbering of the compounds and this creates problems in finding the right structures when referring to for example falcarinol type polyacetylenes etc. later in the text. I have therefore decided to keep the current labelling. However, I have split Figure 5 up into two figures (Figure 5 and 6) so the labeling in these figures more or less follow the compound labelling in the text, which also make sense since they constitute a miscellaneous group of polyacetylenes that cannot be categorized as the other polyacetylenes.

Comment 2:

“The author does not explain in detail the biological molecular targets of these compounds, only a few sentences about ABCG2 and ALDHs. It would benefit for progressing in the field by determining the potential targets in cells and/or a more descriptive mechanism of action. Does the author not cover this because of the lack of these topics? If they exist, the author should explain in much more detail at the introduction.”

Response to comment 2:

The knowledge about the mechanisms of action and potential targets in cells of acetylenic oxylipins are still not known, although there are some in vitro and in vivo studies that have indicated possible mechanisms of action of the polyacetylenes. Possible mechanisms of actions of acetylenic oxylipins with regard to cell cycle arrest and apoptosis have been described but not in any detail and this is why I do not give a detailed description of this in the introduction. However, in the introduction, I indicate and explain possible mechanisms of action and some of these have more or less also been verified in a few studies as described later in more detail in the text including cell cycle arrest and apoptosis but at the present time as mentioned above we do not know the exact mechanisms of action and targets in cells of these acetylenic oxylipins. The review describe briefly potential mechanisms of action but they need to be studied in more detail in future studies in combination with additional pharmacokinetic studies and in vivo studies in order to understand their main mechanisms of action and targets in cells. This has now been mentioned in the conclusion.

Comment 3

“At some points, the text points out “data no shown” although it will be convenient to change for a statement “see reference X for a fully description of derivatives of X which are not included because of poor activity” or a legend or something similar. The first time looks weird to the reader although the following ones it is clear why is indicated this.”

Response to comment 3:

Thank you very much for this very good comment. I fully agree and I have now explained in the revised manuscript why the data are not shown as suggested.

Comment 4:

In the figure 2 for the Japoangelol derivatives, from a chemical point of view it makes no sense the drawing and it does not specify with is the atom of A/B conjugated to the oxylipin scaffold. It shall be correctly drawn and place the atom bond of the aromatic moieties with a zipped line or similar.

Response to comment 4:

Thank you very much for this very important comment. Fully agree that the drawing does not make any sense from a chemical point of view. The drawing of the Japoangelol derivatives has been corrected in the revised manuscript so they now make sense.

Regarding spelling errors:

All spelling errors listed below have been corrected in the revised manuscript. Furthermore, the whole manuscript has been checked for further spelling errors and a few additional have also been corrected.

Line 134, “indicates” by “indicate”

Line 277, shall be written cytotoxicity

Line 370 “appear” instead “appears”

Line 398 “ntis” acronym of ?

Line 466, shall be written pro-apoptosis

Line 591 “resonance” and “carbocation”

Line 729 “indicate”

Line 768 “seems”

Reviewer 2 Report

The manuscript molecules-821893 was prepared very carefully and thoroughly. It presents very valuable material. 225 references were cited. All figures were prepared at a high level. Deep SAR analyzes have been made. I strongly recommend this manuscript for publication.

Author Response

The manuscript molecules-821893 was prepared very carefully and thoroughly. It presents very valuable material. 225 references were cited. All figures were prepared at a high level. Deep SAR analyzes have been made. I strongly recommend this manuscript for publication.

I wish to thank the reviewer for these very nice comments to the manuscript.

Reviewer 3 Report

The review authored by Prof. Lars Porskjær Christensen entitled "Bioactive C17 and C18 Acetylenic Oxylipins from Terrestrial Plants as Potential Lead Compounds for Anticancer Drug Development" is clear, well written, precise and easy to follow.

Christensen have summarized what is new in literature about C17 and C18 acetylenic oxylipins from terrestrial plants and their toxicological and pharmacological activities. Molecular mechanisms of action and structure–activity relationship are briefly revised.

The isolates presented glycosylated bonds directly to kaempferol aglycone, mainly galactose and glucose, which established pairs of compounds group. The eight isolates exhibited inhibitory effects against UVB-induced MMP-1 secretion using Hs68 cells, which indicated that C. japonica seeds cake is a promising material for anti-skin aging cosmetics. The undescribed kaempferol is a tetra-glycoside.

Major concerns: (i) This revision is quite broad and should be narrowed down to the some period of time. From which year author did author start considering published data? This should be mentioned and more details about what was/were the keywords author looked for, where (which database)?, and considered as “a positive hit” for his revision.

(ii) Chemical structures showing the C17 and C18 acetylenic oxylipins are clear and precise. However, figures are too simple. Author should add some schemes and make them more attractive. In figure 6, author mentioned “illustration of structural requirements for optimal cytotoxicity”. What did author mean with “structural requirement”? Did you mean reactive group? Is this a new term? Are they reactive sites of the molecule instead? Why not consider reactivity x toxicity? Why not mention their electrophilic or nucleophilic reactivity and possible toxicity, especially for specific groups from biomolecules?

Minor concern: “in vivo” and “in vitro” should be in italic. This manuscript needs to be carefully revised to improve the grammar and readability. Some words are misspelled.

Concluding, this revision is well written, has important chemical message, and might be of interest to researchers who work with phytochemistry and other readers of Molecules. However, in the present form, I recommend minor revision before publication.

Author Response

I thank this reviewer for the positive and constructive comments and suggestions to improve the manuscript. Below is the answer to the comments of this reviewer and changes made in the revised manuscript are indicated.

Comment 1:

“This revision is quite broad and should be narrowed down to the some period of time. From which year author did author start considering published data? This should be mentioned and more details about what was/were the keywords author looked for, where (which database)?, and considered as “a positive hit” for his revision.”

Response to comment 1:

I am not completely sure what is meant by this comment. This review focus on a special group of bioactive C17 and C18 polyacetylenes and their cytotoxic and anti-inflammatory effects to describe their potential as lead compounds for anticancer drug development. The review covers important literature on this topic i.e. as stated in the review from the beginning of 1980’s. The literature search is of course made on all years from the beginning of 1900’s in Web of Science (All databases). I have worked with these compounds for many years and has published articles within the topic so I had in depth knowledge to the literature on the topic before starting writing the review. However, I did of course a literature search with keywords such as “acetylene*”, “polyacetylene*”, “oxylipin*” combined with key words such as “anticancer”, “chemopreventive”, “cytotoxicity”, “cytotoxic”, and “anti-inflammatory” as well as in different combinations in Web of Science (All databases). By using these keywords, it was possible to obtain all relevant literature on the topic. I agree that it sometimes make sense to describe in detail how the literature search is performed (for example in clinical based review articles) but I do not believe that this is relevant for this review, and I have therefore decided not to include this information in the review.

Comment 2:

Chemical structures showing the C17 and C18 acetylenic oxylipins are clear and precise. However, figures are too simple. Author should add some schemes and make them more attractive. In figure 6, author mentioned “illustration of structural requirements for optimal cytotoxicity”. What did author mean with “structural requirement”? Did you mean reactive group? Is this a new term? Are they reactive sites of the molecule instead? Why not consider reactivity x toxicity? Why not mention their electrophilic or nucleophilic reactivity and possible toxicity, especially for specific groups from biomolecules?

Response comment 2:

The figures show the chemical structures of the polyacetylenes discussed in the review and maybe they are simple but they are relevant and cannot be made more advanced. I have not considered to include “advanced” figures of molecular biology models illustrating the production of for example apoptotic biomarkers and how the polyacetylenes may produce cytoprotective proteins by activating the Keap1-Nrf2 pathway etc. because this is not the aim of this review and would make the review unnecessary longer without adding important information.

Regarding the use of the word “Structural requirements” for cytotoxicity, I admit that this is perhaps not very specific, although it has been used in the literature (see for example J. Med. Chem. 2000, 43, 4508-4515). To be more specific, I have therefore changed the title of section 2 from “Structural Requirements for Optimal Cytotoxicity of C17 and C18 Acetylenic Oxylipins” to “Moieties and Stereochemistry that are Important for the Cytotoxicity of C17 and C18 Acetylenic Oxylipins” and changed the text in this section accordingly.

Comment 3:

Minor concern: “in vivo” and “in vitro” should be in italic. This manuscript needs to be carefully revised to improve the grammar and readability. Some words are misspelled.

Response to comment 3:

“in vivo” and “in vitro” has been corrected to italic throughout the manuscript, including references and all misspellings and grammar have been corrected.